# CODEMMLU: A MULTI-TASK BENCHMARK FOR ASSESSING CODE UNDERSTANDING & REASONING CAPABILITIES OF CODELLMS

**Dung Nguyen Manh**[†][∗] **Thang Phan Chau**[†]**, Nam Le Hai**[‡]**, Thong T. Doan**[†]**, Nam V. Nguyen**[†]**, Quang Pham**[◇]**, Nghi D. Q. Bui**[†][∗]
[†]FPT Software AI Center, Viet Nam,
[‡]Hanoi University of Science and Technology,
[◇]Independent Researcher

## ABSTRACT

Recent advances in Code Large Language Models (CodeLLMs) have primarily focused on open-ended code generation, often overlooking the crucial aspect of **code understanding & reasoning**. To bridge this gap, we introduce CodeMMLU, a comprehensive multiple-choice benchmark designed to evaluate the depth of software and code comprehension in LLMs. CodeMMLU includes nearly 20,000 questions spanning diverse domains, including code analysis, defect detection, and software engineering principles across multiple programming languages. Unlike traditional benchmarks that emphasize code generation, CodeMMLU assesses a model's ability to reason about programs across a wide-range of tasks such as code repair, execution reasoning, and fill-in-the-blank challenges. Our extensive evaluation reveals that even state-of-the-art models struggle with CodeMMLU, highlighting significant gaps in comprehension beyond generation. By emphasizing the essential connection between code understanding and effective AI-assisted development, CodeMMLU provides a critical resource for advancing more reliable and capable coding assistants. CodeMMLU is publicly available at: ⬡ CodeMMLU

## 1 INTRODUCTION

Recent advancements in Code Large Language Models (CodeLLMs) (Wang et al., 2021; 2023c; Feng et al., 2020; Allal et al., 2023; Li et al., 2023; Lozhkov et al., 2024b; Guo et al., 2024b; Pinnaparaju et al., 2024; Zheng et al., 2024b; Roziere et al., 2023; Nijkamp et al., 2022; Luo et al., 2023; Xu et al., 2022; Bui et al., 2023; Hui et al., 2024; Bui et al., 2022; Dau et al., 2024b;a) have demonstrated impressive capabilities across various software engineering (SE) tasks (Bui et al., 2022; Dau et al., 2024a; To et al., 2023; White et al., 2024; Sobania et al., 2023; Phan et al., 2024; Sun et al., 2023; Nguyen et al., 2022; Bui, 2019; Zhang et al., 2022; Wang et al., 2023b). However, existing benchmarks often fall short to provide rigorous and reliable evaluations, largely due to outdated methodologies and the risk of data leakage (Matton et al., 2024). Moreover, practical applications of CodeLLMs reveal limitations such as bias and hallucination (Rahman & Kundu, 2024; Liu et al., 2024a) that current benchmarks fail to adequately address.

The predominant focus of coding-related benchmarks has been on open-ended, free-form generation tasks, such as code generation/code completion (Iyer et al., 2018; Lu et al., 2021; Chen et al., 2021; Austin et al., 2021; Lai et al., 2023; Hendrycks et al., 2021; Ding et al., 2023; Zhuo et al., 2024) and other SE tasks like program repair Ouyang et al. (2024); Xia et al. (2023) (Table 1). While appealing, these benchmarks struggle to discern whether CodeLLMs truly understand code or merely reproduce memorized training data (Carlini et al., 2022; Nasr et al., 2023). Additionally, the reliance on test cases and executability for evaluation limits the quantity and diversity of these benchmarks across domains, potentially leading to biased and limited generalizations. Recent efforts to improve

---

[∗]Corresponding authors: Dung Nguyen Manh (`dungnm31@fpt.com`) and Nghi D. Q. Bui (`bdqnghi@gmail.com`).

[†]The author contributed to this work while working with FPT Software AI Center.

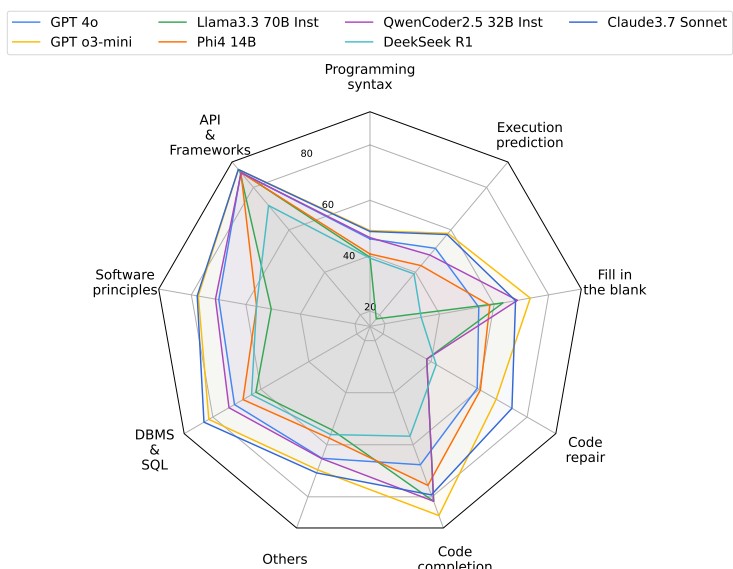

Figure 1: **Summary performance of LLMs on the CodeMMLU benchmark.** This radar chart presents the evaluation results (accuracy %) of different models across various CodeMMLU tasks.

evaluation through free-form question answering (Liu & Wan, 2021; Li et al., 2024) have introduced new challenges, often requiring less rigorous metrics or LLM-as-a-judge approaches (Zheng et al., 2023). However, LLMs-as-a-judge methods are susceptible to adversarial attacks (Raina et al., 2024), raising concerns about the reliability of such evaluation pipelines for coding tasks.

To address the aforementioned shortcomings, we introduce CodeMMLU, a novel benchmark designed to evaluate CodeLLMs' ability to comprehend and reason about code through multiple-choice question answering (MCQ). This approach enables a deeper assessment of how CodeLLMs grasp coding concepts, moving beyond the mere generation capabilities. Inspired by the MMLU dataset (Hendrycks et al., 2020) from natural language understanding, CodeMMLU offers a robust and easy evaluation with the following key features. CodeMMLU comprises nearly 20,000 questions, facilitating a robust and comprehensive evaluation. Its large-scale data curation process mitigates potential biases and improves statistical reliability in measuring CodeLLMs' performance across many capabilities. CodeMMLU covers over 50 software engineering disciplines and more than 10 programming languages, providing a holistic evaluation of CodeLLMs. The MCQ format is highly scalable, allowing for an accurate and straightforward assessment using precision-based metrics. Moreover, by incorporating permutations of answer choices, CodeMMLU emphasizes the model's code understanding capabilities rather than memorizing the training datasets. Consequently, CodeMMLU provides a robust and accurate assessment of the models capabilities to understand software tasks.

CodeMMLU evaluates LLMs' abilities in coding and software problem-solving from a fresh perspective, extending beyond conventional code generation and completion tasks. Our analysis uncovers several key insights: (1) previously unidentified bias issues in CodeLLMs, consistent with those observed in natural language MCQA tasks; (2) proprietary models (GPT-4o and Claude 3.5 Sonnet) consistently deliver the highest average performance; (3) among open-source models, the DeepSeek and Meta-Llama families achieve the greatest accuracy; (4) scaling laws tied to model size hold partially within the same model family but not across families, highlighting the critical roles of pre-training, post-training, and model architecture; (5) advanced prompting strategies, such as Chain-of-Thought (CoT), consistently impair performance, casting doubt on CodeLLMs' reasoning capabilities for complex, multi-step tasks; and (6) when code completion benchmarks (e.g., HumanEval) are reframed from open-ended generation to MCQA format, LLMs exhibit reduced performance, questioning their true understanding of code. Notably, even strong reasoning models like DeepSeek-R1 underperform on CodeMMLU, suggesting significant room for improvement in

future iterations. These findings highlight the shortcomings of CodeLLMs in truly understanding code and reveal several promising future research avenues.

In summary, this work makes the following contributions:

1. We present the first MCQ benchmark for software and coding-related tasks, addressing the need for a comprehensive and large-scale evaluation in the code domain. CodeMMLU enables the evaluation of LLMs' alignment with human inference in the software knowledge domain, similar to advances in the NLP field.

2. CodeMMLU provides a thorough assessment of LLM capabilities, ensuring a substantial number of samples and the diversity across tasks, domains, and languages. This enables a more nuanced understanding of an LLM's strengths and weaknesses, facilitating the development of models better aligned with the complexities and demands of the software domain.

3. Our experiments offer critical insights into LLM performance, highlighting the impact of factors such as model size, model family, and prompting techniques. Notably, CodeMMLU unveils a performance gap between LLMs' code generation and comprehension abilities. Additionally, we identify LLMs' sensitivity to the selection biases introduced by the MCQ format. These findings provide valuable guidance for the community to enhance the robustness, adaptability, reliability, and domain-specific capabilities of LLMs in real-world software engineering.

## 2 RELATED WORK

**Code Intelligence Benchmarks.** The rapid development of Large Language Models (LLMs) for code-related tasks necessitates the development of diverse benchmarks to evaluate their performance. Algorithm-focused benchmarks, such as HumanEval (Chen et al., 2021), MBPP (Austin et al., 2021), and their extended versions (e.g., HumanEval+, MultiPL, MBPP+) (Liu et al., 2024b), focus on small-scale code generation tasks but lack the depth needed to assess broader comprehension. More challenging tasks, such as those in CodeContests (Li et al., 2022) and LiveCodeBench (Jain et al., 2024), provide competitive programming problems but remain primarily generative (Table 1). Comprehensive evaluation frameworks, such as CodeXGLUE (Lu et al., 2021), XLCoST (Zhu et al., 2022), and XCodeEval (Khan et al., 2023), provide versatility through multi-task assessments. However, these benchmarks are either dependent on metrics like BLEU and ROUGE, or testcase execution, which limits their reliability and scalability for a large-scale, comprehensive evaluation of LLM.

In contrast, multiple-choice question (MCQ) benchmarks offer a more standardized, scalable, and reliable evaluation method, as demonstrated in popular general-purpose benchmarks such as MMLU (Hendrycks et al., 2020) and TruthfulQA (Lin et al., 2022). Although MCQs facilitate large-scale assessments, recent studies highlight their susceptibility to biases, such as sensitivity to the answer choice orders (Wang et al., 2023a; Robinson et al., 2023). Existing MCQ benchmarks also lack focus on software engineering, limiting their applicability to code-related evaluations. In contrast, we curated data from a wide range of tasks and applied various filtering and debiasing techniques to improve CodeMMLU's comprehensiveness, while minimizing data leakage and biases.

**Understanding & Reasoning on Code** There is a large body of research leveraging AI models for reasoning about code Gu et al. (2024a); Chen et al. (2024); Liu & Jabbarvand (2025); Dehghan (2024); Le et al. (2024b;a); Bieber et al. (2020); Shi et al. (2019); Li et al. (2021). Reasoning about code requires AI models to comprehend both its syntactic and semantic aspects. In the early days, static analysis was the primary approach to reasoning about code. With the rise of deep learning, methods that model code structures Mou et al. (2016); Bui et al. (2021b;a) emerged as promising alternatives. Subsequently, graph-based representations of code have gained traction, enabling reasoning about program properties and execution using graph neural networks (GNNs) Le et al. (2024a); Bieber et al. (2020); Shi et al. (2019). With the advent of large language models (LLMs), these models have been directly applied to reasoning about program behavior in downstream tasks Gu et al. (2024a); Chen et al. (2024); Liu & Jabbarvand (2025); Dehghan (2024); Le et al. (2024b). CodeMMLU provides a comprehensive benchmark that reflects real-world reasoning requirements by offering ground-truth answers to programming-related questions, allowing LLMs to derive final solutions through various reasoning paths.

Table 1: **Comparison between common code understanding benchmarks for LLMs in terms of coverage of foundation tasks of programming comprehension model.**

| Benchmark | Question | | SWE knowl-edge | Programming Task | | | Test size |
|---|---|---|---|---|---|---|---|
| | Open-end | MCQ | | Code composi-tion | Code compre-hension | Code debug-ging | |
| APPS Hendrycks et al. (2021) | ✓ | | | ✓ | | | 5000 |
| MBPP Austin et al. (2021) | ✓ | | | ✓ | | | 974 |
| HumanEval Chen et al. (2021) | ✓ | | | ✓ | | | 164 |
| CRUXEval Gu et al. (2024b) | ✓ | | | | ✓ | | 800 |
| LiveCodeBench Jain et al. (2024) | ✓ | | | ✓ | ✓ | ✓ | 880[1] |
| CodeApex Fu et al. (2023) | ✓ | ✓ | ✓ | ✓ | | ✓ | 2.056 |
| **CodeMMLU** | | ✓ | ✓ | ✓ | ✓ | ✓ | **19.912** |

# 3 CODEMMLU: DATA CURATION

The CodeMMLU benchmark is structured into two primary categories: (i) knowledge-based tests, designed to evaluate programming knowledge through questions addressing both syntactic and semantic aspects, and (ii) fundamental coding-skill tests, created by transforming high-quality codebase seeds into task-specific challenges. CodeMMLU includes nearly 20,000 questions spanning 52 diverse topics (Table 2). We design the knowledge-based tests to probe multi-level cognitive structures, assessing an LLM's understanding of software knowledge at both semantic and syntactic levels. In contrast, the fundamental coding-skill test sets align with the cognitive process model of Shneiderman & Mayer (1979), focusing on core programming tasks that mimic real-world problem-solving scenarios.

## 3.1 KNOWLEDGE-BASED TASK CREATION

The knowledge-based test sets are designed to cover a wide range of topics and follow the multi-level cognitive structures model (Shneiderman & Mayer, 1979) which combines syntactic and semantic knowledge. The subset target is to measure the LLM's coding capability and comprehensibleness of programming concepts. We collected raw programming-related questions and their corresponding multiple-choices answer from W3School (W3Schools, 2024) and Common Crawl project[2] (See more license detail in Appendix A.3). The knowledge-based test set include:

- **Syntactic subset.** Focused on programming language grammar and structural correctness, such as condition statement, format of iteration, common library usage.

- **Semantic subset.** Targeted more abstract programming concepts, such as algorithms, data structures, object-oriented principles.

We maintain a high-quality evaluation set by filtering the raw data that undergoes a rigorous formatting and deep-learning-based filter in which we remove any instances that do not meet our quality criteria (see in section 3.3 and Appendix A.1). Resulting in an evaluation set (Table 2) that contains more than 11,000 instances, lying in 52 topics classified to 5 main subjects (categorized by source tag).

## 3.2 FUNDAMENTAL TEST CONSTRUCTION

Our benchmark encompasses four distinct MCQ programming tasks designed to assess the foundational capabilities outlined in the cognitive process model of programmer comprehension, namely: composition, comprehension, debugging, and modification.

**Code Completion** evaluates a model's composition ability by requiring it to complete partially written code based on provided requirements. We adapted HumanEval (Chen et al., 2021), originally designed for code generation, into an MCQ format. From its 164 unique programming problems, we employed Large Language Models (LLMs) to generate plausible but incorrect solutions as distractors. All options, including correct solutions migrated from HumanEval and generated incorrect ones, were tested for executability. Some incorrect solutions were designed to pass certain test cases but fail

---

[1]https://github.com/LiveCodeBench/LiveCodeBench
[2]https://commoncrawl.org/

Table 2: **Summary of CodeMMLU Subject Categories and Task Distribution.**

| | Subject | Topic | Source | Testsize |
|---|---|---|---|---|
| *Syntactic knowledge* | API & Frameworks usage | Jquery, Django, Pandas, Numpy, Scipy, Azure, Git, AWS, svg, xml, Bootstrap, NodeJS, AngularJS, React, Vue. | W3Schools, Geeksforgeeks, CommonCrawl | 740 |
| | Programming language syntax | C, C#, C++, Java, Javascript, PHP, Python, R, Ruby, MatLab, HTML, CSS, TypeScript. | | 6,220 |
| *Semantic knowledge* | DBMS & SQL | DBMS, MySQL, PostgreSQL, SQL. | | 393 |
| | Software principles | Data structure & Algorithm, Object-oriented programming, Compiler design, Computer organization and Architecture, Software Development & Engineering, System Design. | | 3,246 |
| | Others | Program accessibility, Computer networks, Computer science, Cybersecurity, Linux, Web technologies, AWS. | | 1,308 |
| *Fundamen-tal task* | | Code completion | HumanEval | 163 |
| | | Fill in the blank | LeetCode | 2,129 |
| | | Code repair | QuixBugs | 76 |
| | | Execution Prediction | IBM CodeNet | 6,006 |

others, adding complexity and challenging models to distinguish between correct and nearly-correct solutions based on semantic and syntactic understanding.

**Code Repair** assesses a model's debugging capability by requiring it to identify and fix errors in provided code snippets. We built this task upon QuixBugs (Lin et al., 2017), which was originally designed for debugging algorithmic programs. We used a "diff" operation on buggy and corrected versions in QuixBugs (Python and Java) to identify specific fixes, which served as correct solutions. To create plausible distractors, we targeted components frequently involved in bugs (e.g., return statements, loop conditions, if/else/switch expressions) and guided LLMs to generate alternative fixes. These alternatives were designed to seem plausible but not fully resolve the bug. Each distractor was verified for incorrectness, and all options were made executable to ensure that models needed a deep understanding of the code to identify and apply the correct fix.

**Execution Prediction** evaluates a model's ability to identify and understand defects within code snippets, focusing on both logical and syntactical errors. This task measures the comprehension and debugging capabilities of LLMs by requiring them to predict the execution outcome of given code. It includes two sub-tasks: detecting any defects/flaws in the provided code and comprehending the output of a certain test sample. We derived this task set from IBM CodeNet (Puri et al., 2021), a large-scale benchmark for algorithmic coding tasks. We focused on Python and Java subsets, collecting both accepted and buggy versions of code. After filtering out duplicates, we created a diverse set of code samples. For each snippet, we provide the correct execution result (golden answer) and three distracting options, which could be one of several possible outcomes: (i) Compile Error, (ii) Time Limit Exceeded, (iii) Memory Limit Exceeded, (iv) Runtime Error, or (v) No abnormally found.

**Fill in the Blank** evaluates a model's code comprehension ability by requiring it to complete missing parts of a code snippet, given documentation and an incomplete code sample. This task assesses not only the model's ability to fill gaps but also its understanding of both high-level programming concepts and low-level grammatical structures. We collected approximately 2,000 coding problems from LeetCode [3], covering solutions in three widely-used programming languages (Python, Java, C++). From each problem's solution, we parsed and randomly selected key components (i.e. crucial elements of the program's logic and flow like loop conditions, expression statements, conditional statements) to be blanked out. To create plausible but incorrect options for the multiple-choice question (MCQ) format, we employed LLM to generate alternative solutions for the blanked-out components. These distractors were designed to be contextually relevant but incorrect, adding complexity to the task. We executed all generated options to verify their incorrectness, ensuring they do not solve the problem as intended.

---

[3]https://leetcode.com/

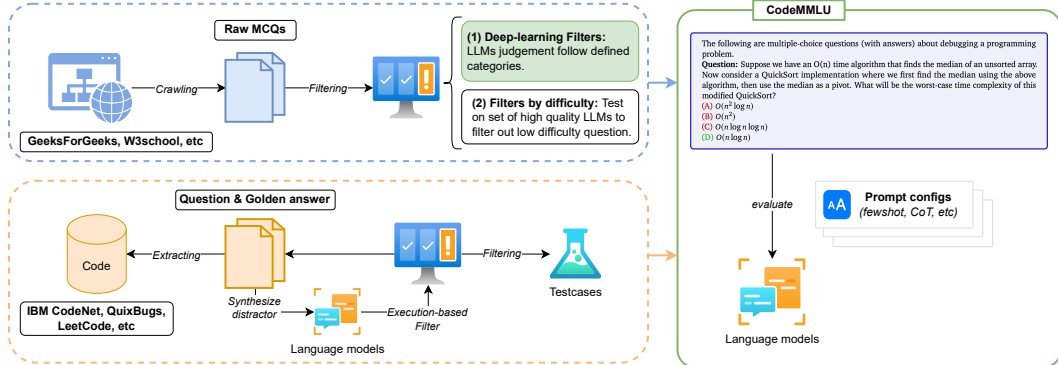

Figure 2: **Overview of CodeMMLU data creation pipeline.** The blue diagram describe the process of collecting raw multiple-choice questions (MCQs) from open source internet for a knowledge testset. Otherwise, the pipeline of real-world problem indicated in orange area.

## 3.3 DATA CLEANING

The preprocessing process (described in Figure 2) includes a deep learning-based filtering and execution-based filtering to ensure that each question met the desired quality standards, including clarity, lack of ambiguity, and difficulty.

**LLM-based Filtering** To begin, we employed an LLM-based filter to assess the instances in the crawled knowledge test set. Each instance was evaluated based on three criteria: Completeness, Coherence and clarity, and Coding relevance. The models utilized for this evaluation included GPT-3.5, Llama3.1-8B Instruct, and Mixtral-8×7B Instruct. We averaged the scores and used them to select a filtering threshold for each criteria (see discussion in Appendix A.1). To detect and handle duplications, we applied the MinHash LSH algorithm Zhu et al. (2023), configured with 256 permutations, to cluster near-duplicate questions. We remove all false positive instances in each cluster with 0.8 as the similarity threshold. To verify the efficacy of the LLM-based filter, we randomly selected 100 instances from each subject area for manual verification against the three criteria.

**Execution-based Filtering** To ensure the question correctness, we apply an execution-based filtering in the fundamental test sets. We merge the distractor of (i) code completion, (ii) fill-in-the-blank, and (iii) code repair with their codebase and execute with their corresponding test cases. The distractor is designed to bring challenge since it requires LLM to comprehend their correctness without executing it, we select distractors that are executable with 0 to few (less than 50%) test cases passed in their execution result. In the other hand, the task Execution Prediction's groundtruth are collected from executing process, the distractor are randomly pick from common executing scenarios.

## 4 EXPERIMENTAL RESULTS

### 4.1 SETUP

**Model selection.** We evaluate CodeMMLU on 40 popular open-source LLMs, covering a wide range of parameter sizes and architectures. The models were selected from 13 different families, with parameters ranging from 1 billion to over 70 billion. Each family included base and instructed/chat versions. In addition to open-source models, we also included several proprietary models from OpenAI and Anthropic to ensure a comprehensive coverage of the state-of-the-art in language modeling. All model information can be found at C.

**Answer extraction.** CodeMMLU leverages the MCQ format for scalability and ease of evaluation. In order to maintain this advantage, we only apply simple regex methods to extract the selection answer (i.e., extract by directly answering (A|B|C|D) or containing the pattern "answer is A|B|C|D"). The model response is required to be parsable; otherwise, it will be marked as unanswered.

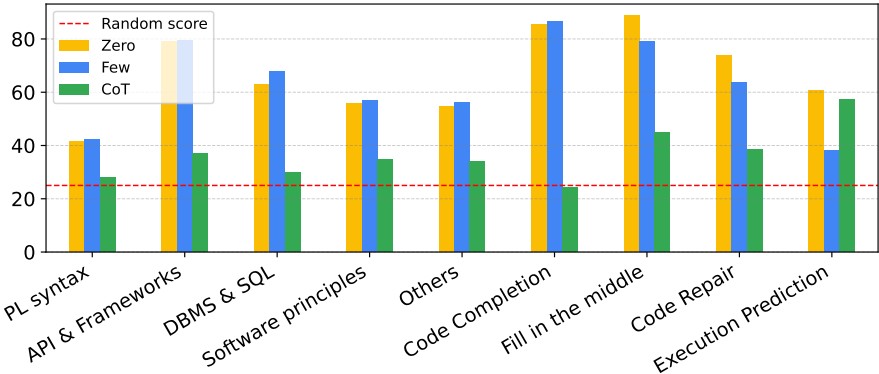

Figure 3: **Comparison of prompt configuration on GPT-4o.** The experiment exposes the drawback of Chain-of-Thought prompting technique in term of boosting performance on task that not require logic or reasoning.

In the following, we present key findings of CodeMMLU on (i) knowledge and fundamental test correlation; (ii) MCQ bias evidance; (iii) Disagreement between code-generation alike benchmark and MCQ format. Due to space constraints, we provide detail experimental results, additional discussions and analyses in the appendix, including assessing data leakage (Appendix A.2); MCQs analysis (Appendix B.1); Chain-of-thought technique analysis B.2 and full 43 LLMs results B.3.

## 4.2 KEY INSIGHTS

**Overall performance** CodeMMLU revealed significant performance differences across models, as shown in Table 3. OpenAI's GPT-4o outperformed all models on CodeMMLU, demonstrating its quality across diverse tasks (Figure 1). Notably, despite not being the latest model, the instructed version of Llama3.1 70B from Meta achieved the highest score among open-source models from 13 families. While LLMs perform well on knowledge-based tasks, they struggle with real-world problems, particularly in execution prediction tasks (see Appendix B.3 for all the experiment details.)

Figure 4 illustrates CodeMMLU's capability to measure LLMs' coding knowledge and skills across a wide range of subjects. Our benchmark provides clear, distinct rankings that establish a higher hierarchy of models compared to other benchmarks (see the result in Table 7). Interestingly, the results do not strictly adhere to scaling laws (Kaplan et al., 2020), where larger parameter sizes typically outperform smaller ones. This highlights the impact of data quality in the LLM pretraining process, as recently released models often achieve comparable performance to larger models from previous versions. CodeMMLU also indicates the importance of instruction tuning in improving model performance on complex tasks. Models with instruction tuning substantially outperform their non-instructed counterparts, for example, DeepSeek-Coder-33b surpasses its base model by approximately 29%.

**The impact of model reasoning and Chain-of-Thought (CoT) prompting** Although CoT prompting (Wei et al., 2023) is often expected to enhance performance by eliciting deeper reasoning, our experiments reveal that CoT and reasoning models may not always offer improvements in CodeMMLU. Table 3 shows that DeepSeek R1 performs significantly worse than its base model, DeepSeek V3, despite being designed for reasoning tasks. Meanwhile, GPT o3-mini, a native reasoning model, achieves the best results. Our analysis suggests that, apart from GPT o3-mini, other reasoning models (e.g, GPT o1, DeepSeek R1) tend to overreason across all CodeMMLU tasks. In contrast, o3-mini demonstrates the ability to decide when to apply reasoning versus directly answering knowledge-seeking tasks, such as in syntactic and semantic evaluations (Figure 11 12 ). These findings align with Chen et al. (2025), which observes that current reasoning models often overreason even on simple questions. This suggests the need for a more effective decision-making mechanism to determine when reasoning is beneficial. Table 6 further highlights that while R1 generates the longest responses (in tokens), it underperforms compared to the latest top-tier LLMs.

We further investigate the impact of different prompting techniques and report the results in Figure 3. The results show a significant decline in GPT-4o's performance with CoT, suggesting that the additional complexity introduced by step-by-step reasoning does not align well with knowledge-seeking tasks (see Appendix B.2 for more discussions). In contrast, few-shot prompting consistently emerges as the most reliable and effective strategy across various tasks, offering a balanced approach without overwhelming the models. Overall, with the exception of GPT o3-mini, we found no improvements from reasoning models or CoT prompting, suggesting that CodeMMLU presents a challenging benchmark to test the models reasoning capabilities in code domains.

Table 3: **Summary of LLM Family Performance on CodeMMLU**. The evaluation results (accuracy %) of different language models across the CodeMMLU task (CodeMMLU column represents the accuracy average among all subject).

| Family | Model name | Size (B) | Knowledge test | | Fundamental test | CodeMMLU |
| | | | Syntactic | Semantic | | |
| --- | --- | --- | --- | --- | --- | --- |
| | | | Closed-source models | | | |
| Claude | Claude3.7 Sonnet | - | 52.78 | **76.26** | *60.92* | *61.65* |
| | Claude3.5 Sonnet | - | 52.23 | 73.45 | 58.56 | 59.81 |
| | Claude3.5 Haiku | - | 49.24 | 68.20 | 57.83 | 57.25 |
| | Claude3 Sonnet | - | **67.22** | 66.08 | 38.26 | 53.97 |
| GPT | GPT o3-mini | - | 53.08 | *75.50* | **62.77** | **62.36** |
| | GPT 4o | - | 50.63 | 69.61 | 53.89 | 56.40 |
| | GPT 4o-mini | - | 48.66 | 55.90 | 20.33 | 38.43 |
| | GPT-3.5-turbo | - | *61.68* | 53.65 | 45.26 | 51.70 |
| | | | Open-source models | | | |
| Llama | Llama3.3 70B Inst | 70 | 44.31 | 52.76 | 30.96 | 40.66 |
| | Llama3.1 405B Inst | 405 | 50.82 | **71.41** | **57.10** | **58.23** |
| | Llama3 70B Inst | 70 | 46.94 | 62.64 | 53.15 | 53.19 |
| | CodeLlama34B Inst | 34 | 56.81 | 46.93 | 23.55 | 38.73 |
| DeepSeek | DeepSeek R1 | 671 | 42.39 | 56.77 | 38.08 | 43.91 |
| | DeepSeek V3 | 685 | 48.30 | 57.53 | 45.06 | 49.08 |
| | DeepSeekCoder 33B Inst | 33 | 53.65 | 45.43 | 21.46 | 36.60 |
| | DeepSeekMoE 16B Chat | 16.4 | 31.74 | 35.42 | 27.32 | 31.01 |
| Mistral | Mistral7B Inst (v0.3) | 7 | 54.42 | 51.25 | 31.85 | 43.33 |
| | Mixtral 8×7B Inst | 46.7 | *61.17* | 54.89 | 24.09 | 42.96 |
| | Codestral 22B | 22 | 60.34 | 52.10 | 37.85 | 47.60 |
| Phi | Phi4 | 14 | 45.34 | 57.46 | 47.82 | 49.19 |
| | Phi4 Mini Inst | 12 | 41.94 | 51.59 | 19.75 | 34.85 |
| Qwen | Qwen2.5 14B Inst | 14 | 46.38 | 58.70 | 51.49 | 51.38 |
| | QwQ 32B Preview | 57 | **61.34** | 57.48 | 30.48 | 46.34 |
| | QwenCoder2.5 32B Inst | 32 | 50.63 | *69.61* | *53.89* | *56.40* |

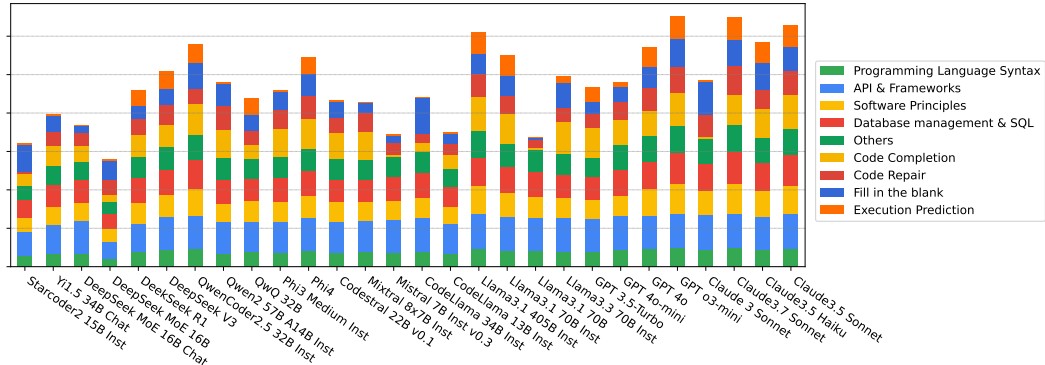

Figure 4: **CodeMMLU accuracy by task on LLMs.** While knowledge tasks are following the scaling law, real-world tasks offer more challenges to LLMs which indicate the performance of instruction tuning and data quality when evaluating on CodeMMLU.

**Correlation Between Software Knowledge and Real-World Performance**   Our experiments revealed a strong correlation between performance on knowledge-based tasks and real-world coding challenges. Specifically, the Pearson correlation score r = 0.61 between model rankings on the knowledge test set and their performance on real-world problems, derived from the accuracy of 43 LLMs across 15 model families, indicates a moderate alignment (Figure 5). This suggests that models demonstrating a deeper understanding of software principles consistently excel in real-world coding tasks, highlighting the importance of foundational knowledge for practical coding performance.

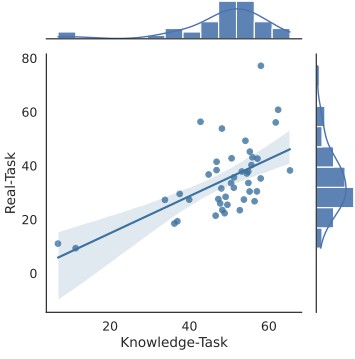

Figure 5: **Correlation between knowledge tests and fundamental skill tests.** Experiments on 10 LLM families show a clear alignment between models with a strong understanding of software knowledge and their performance on diverse problem-solving tasks in the CodeMMLU fundamental skill tests.

Figure 6: **Comparison Between LLM Response Length and CodeMMLU Score.** While reasoning models (e.g., DeepSeek-R1) generate longer, reasoning-heavy responses, this does not necessarily correlate with higher accuracy on CodeMMLU.

**Selection bias in MCQs format**   We experimented with multiple answer order permutations (follow Zheng et al. (2024a)), the result displayed significant inconsistent behavior exhibited by LLMs when swapping golden answer positions. As presented in Table 7, the model's performance changes dramatically in each answer order configuration, which is based on the correct answer's position. The LLM's accuracy fluctuates between different permutations (i.e. DeepSeek-Coder-34B $\Delta\sigma = 36.66$), demonstrating how sensitive it can be to the structure and order of answers (Figure 9). However, Table 8 indicate the different of MCQ bias between strong models (e.g GPT-4o, Claude3-orpus) and others, which highlight the consistency and robustness among them (see discussion in B.1).

**Disagreement between Open-ended generation benchmark and MCQ Code completion**   A notable finding from our experiments is the discrepancy in model performance between open-ended benchmarks and multiple-choice formats. Specifically, when comparing the original HumanEval questions with their multiple-choice equivalents in our CodeMMLU code completion set, we found that models performing well on HumanEval do not consistently replicate their success in CodeMMLU. For instance, when evaluating identical questions across the formats, the number of cases where models answered both correctly or incorrectly was unexpectedly low. The correlation scores in Figure 7 further illustrate the weak alignment of success between these two benchmarks, revealing that performance in open-ended tasks does not reliably predict performance in multiple-choice coding tasks. This lack of alignment suggests that traditional benchmarks might overestimate a model's understanding by focusing too narrowly on code generation, which is highly susceptible to data leakages. In contrast, CodeMMLU requires the models to engage in complex reasoning to understand code and solve software engineering problems.

## 5   CONCLUSIONS

In this work, we introduced CodeMMLU, a comprehensive and scalable benchmark designed to evaluate large language models' (LLMs) capabilities across a wide range of software knowledge

Table 4: **Performance Comparison between HumanEval and MCQ Code Completion Tasks.** The performance fluctuation highlights the selection biases observed when the correct (golden) answer is moved to positions A, B, C, or D.

| Models | HumanEval | Code Completion MCQ | | | |
|---|---|---|---|---|---|
| | | A | B | C | D |
| CodeLlama-7B-Python | 40.48 | 0.00 (-40.48) | 90.24 (+49.76) | 14.02 (-26.46) | 0.61 (-39.87) |
| CodeLlama-7B-Instruct | 45.65 | 3.66 (-41.99) | 1.22 (-44.43) | 93.90 (+48.25) | 15.85 (-29.80) |
| CodeLlama-13B-Python | 42.89 | 0.61 (-42.28) | 54.88 (+11.99) | 70.12 (+27.23) | 12.20 (-30.69) |
| CodeLlama-13B-Instruct | 50.6 | 2.44 (-48.16) | 68.29 (+17.69) | 72.56 (+21.96) | 29.88 (-20.72) |
| CodeLlama-34B-Python | 45.11 | 0.61 (-44.50) | 77.44 (+32.33) | 70.73 (+25.62) | 49.39 (4.28) |
| CodeLlama-34B-Instruct | 50.79 | 9.15 (-41.64) | 84.76 (+33.97) | 65.24 (+14.45) | 46.34 (-4.45) |
| Deepseek-Coder-7B-base-v1.5 | 43.2 | 40.85 (-2.35) | 74.39 (+31.19) | 64.02 (+20.82) | 39.02 (-4.18) |
| DeepSeek-Coder-33B-base | 56.1 | 1.22 (-54.88) | 82.32 (+26.22) | 75.00 (+18.90) | 56.10 (0.00) |
| Phind-CodeLLama-34B-v2 | 71.95 | 6.10 (-65.85) | 90.85 (+18.90) | 75.00 (+3.05) | 46.34 (-25.61) |
| Mixtral-8x7B-Instruct-v0.1 | 40.2 | 22.56 (-17.64) | 74.39 (+34.19) | 71.95 (+31.75) | 63.41 (+23.21) |

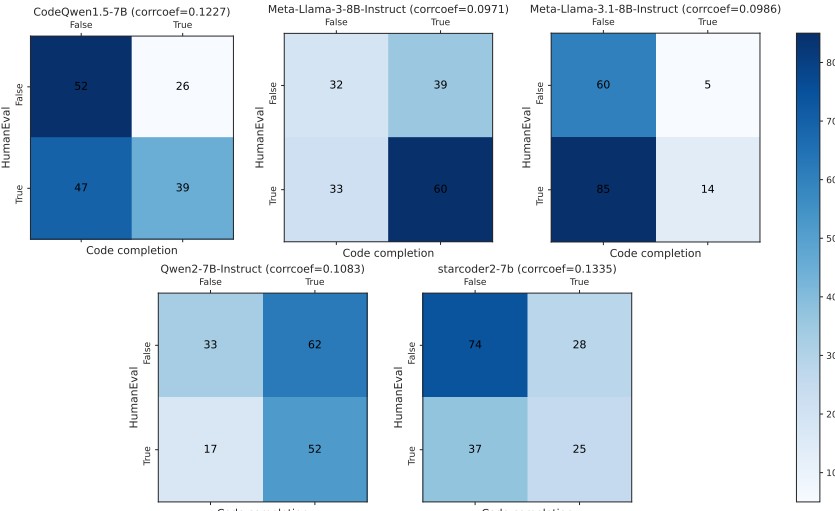

Figure 7: **Comparison of CodeMMLU's code completion task and HumanEval.** Many LLMs show a performance discrepancy between the two tasks, where models that successfully passed the HumanEval code generation test often failed to select the correct answer in the multiple-choice (MCQ) format, or vice versa, for the same question.

and real-world programming tasks. Our experiments highlighted the benchmark's key advantages, including its cost-effectiveness, scalability, and extensive task coverage. The insights gained revealed a strong correlation between software knowledge and real-world task performance, demonstrating that models with deeper comprehension outperform those relying purely on probabilistic generation.

Additionally, CodeMMLU provides more accurate and detailed rankings of LLMs, particularly in open-source models, where significant reordering of performance was observed. The benchmark also revealed inconsistencies in model comprehension when compared to traditional evaluations like HumanEval, emphasizing the need for more robust benchmarks that go beyond simple code generation.

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

# A   DATASET

## A.1   DATA CLEANING

**Rule-based filtering**   We prefer questions that contain code when collecting data; therefore, MCQs often contain noisy patterns and low-quality questions. In the cleaning process, we defined a heuristic rule-based filter to eliminate incomplete data and non-textual content. First, we detect and eliminate non-textual questions by filtering questions that contain hrefs, image URLs, links to other questions or media. We also applied BeautifulSoup to remove unwanted HTML tags.

**Deep learning-based filtering**   To ensure the CodeMMLU is fully targeted on coding and software-related task, we employed models from OpenAI (GPT-3.5-turbo), Mistral (Mixtral $8\times7$B Instruct), and Llama (Llama3.1 8B) as our annotators to judge the triple criteria: **Completeness**; **coherence and clarity**; and **coding relevance** (check appendix C.1 to see the prompt). We averaged LLM ratings by category and selected a threshold of 4 in 3 aspects. Result of removing $\approx 25.6\%$ of raw data. On the other hand, we simultaneously sampled a subset of 100 instances in each subject to update our filter rule. (Figure 8

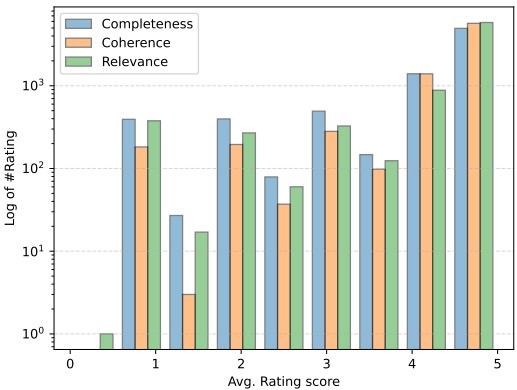

Figure 8: **LLM-based filter score distribution.**

**Execution-based filtering**   After synthesizing the fundamental task's distractor (i.e., false answer), we concatenated and executed them as a complete function in an isolated environment. The code completion and fill-in-the-blank tasks have their original test cases, while code repair needs to synthesize new test cases. Therefore, we extracted the method signature (using the code-text parser toolkit from Manh et al. (2023)) and synthesized the function input, which later was executed to create test cases for the corresponding function. We ran in parallel the distractor executing on the testcase and synthesized a new distractor, ensuring the distractor collection is executable and able to pass 0-50% test cases.

## A.2   DATA CONTAMINATION

The development of large language models (LLMs) often involves crawling data from diverse sources across the internet, with limited transparency regarding their preprocessing. Given the vast and often proprietary nature of these training datasets, it is widely acknowledged that creating a fully leakage-free benchmark is virtually impossible. While recent benchmarks have recognized this issue and generally accept that avoiding data leakage entirely is extremely difficult, one common mitigation strategy involves filtering data based on its timeline Jain et al. (2024); Wang et al. (2024); Zhang et al. (2023).

In our efforts to address this challenge, we acknowledge the complexity of completely eliminating data leakage. To enhance the reliability of CodeMMLU, we adopt proactive measures during the data creation process. Specifically, we transform seed data into multiple-choice question formats and introduce synthetic distractors. For tasks like code repair and execution prediction, the test sets were extracted from codebase seeds and modified to align with specific task requirements.

To further assess and quantify potential data leakage, we employ the methodology outlined in Xu et al. (2024). This includes calculating *perplexity* and conducting *n-gram* analysis on several well-known models from diverse families (e.g., Mistral, DeepSeek, Llama). The results, presented in Tables 5 and 6, highlight a significant margin between CodeMMLU and other coding benchmarks, reinforcing the reliability of CodeMMLU as a robust evaluation tool.

**Perplexity** measures the uncertainty of a language model when predicting the next token in a sequence Jelinek et al. (1977). Therefore, as low as the perplexity score indicates, the model is confident in predicting the evaluating sequence and the more likely that the model was encountered during the training process. Perplexity is expressed as the exponentiated average negative log-likelihood of a sequence:

$$\text{PPL}(\mathbf{X}) = \exp\left(-\frac{1}{t}\sum_{t=0}^{t}\log p_\theta(x_i|x_{<i})\right) \tag{1}$$

where $\mathbf{X} = [x_0, x_1, ..., x_t]$ denotes a tokenized sequence.

**N-gram Accuracy** Xu et al. (2024) is a metric designed to detect fine-grained data leakage at the instance level by combining the question and answer into a single text (X), uniformly sampling starting points, and predicting the next n-grams based on the given prompts. If most n-grams are accurately predicted, it suggests the model may have encountered the data during training. The N-gram accuracy can be expressed as:

$$\text{N-gram Accuracy}(X) = \frac{1}{S \cdot K}\sum_{i=0}^{S}\sum_{j=0}^{K} I(X_{start_j:start_j+n}, \hat{X}_{start_j:start_j+n}), \tag{2}$$

where $S$ is the dataset size, $K$ is the number of sampled starting points, $X_{start_j:start_j+n}$ is the actual n-gram, $\hat{X}_{start_j:start_j+n}$ is the predicted n-gram, and $I$ checks for exact matches. Author add ROUGE-L and edit distance similarity to provide robustness for augmented datasets. A high accuracy for each n-gram in a prediction indicates a strong likelihood that the sample was seen during the training process. Xu et al. (2024)

Table 5: **Perplexity score comparison between coding benchmark.** *(higher is better)*

| Models | CodeScope | CodeApex | CodeMMLU |
|---|---|---|---|
| Mistral7B-v0.3 | 9.32 | 16.08 | 16.32 |
| DeepSeekCoder7B-v1.5 | 5.26 | 9.39 | 57.36 |
| DeepSeekV2-Lite | 6.89 | 11.99 | 1419.48 |
| Llama-3.1-8B | 10.05 | 123.20 | 197.31 |

Table 6: **5-gram accuracy comparison between coding benchmark.** *(lower is better)*

| Models | CodeScope | CodeApex | CodeMMLU |
|---|---|---|---|
| Mistral7B-v0.3 | 0.2510 | 0.1702 | 0.1365 |
| DeepSeekCoder7B-v1.5 | 0.2818 | 0.1680 | 0.1416 |
| DeepSeekV2-Lite | 0.2492 | 0.1587 | 0.0687 |
| Llama-3.1-8B | 0.2219 | 0.1309 | 0.0652 |

## A.3 LICENSE

In the construction of CodeMMLU, we collect only the multiple-choice questions, problem descriptions, code solutions, and test cases from the publicly visible parts of W3School and Geeksforgeeks quizzes/puzzles and LeetCode. We avoid any data collection that requires login or interaction with these websites. On one hand, most of our knowledge test set ($\approx 61\%$) are collected from Common

Crawl (from portion tagged CC-MAIN-2021-41 to CC-MAIN-2024-46). On the other hand, the fundamental tasks were created on a permissively licensed codebase, namely IBM Project CodeNet (Apache 2.0), HumanEval, QuixBugs (MIT). For data crawled from websites such as W3Schools (fair use for research purposes) and GeeksforGeeks (under the Copyright Act 1957), we fully complied with their copyrights or sought their permission to use such data for this project. CodeMMLU will be published and distributed under the MIT license.

# B ALL EXPERIMENTAL RESULTS

Table 7: **CodeMMLU and other coding benchmarks comparison.** The ranking reorder comparison between CodeMMLU (CM) and other benchmarks (namely HumanEval (HE)).

| Family | Model | Size (B) | MMLU | GSM8k | HumanEval | MBPP | CodeMMLU | HE→CM |
|---|---|---|---|---|---|---|---|---|
| *Closed-source models* | | | | | | | | |
| Anthropic | Claude-3 Sonnet | - | **88.70** | **96.40** | **92.00** | 76.6 | 55.48 | **1**→4 |
| OpenAI | GPT-4o | - | **88.70** | 95.80 | 90.20 | **81.4** | **64.96** | **2**→**1** |
| | GPT-3.5-turbo | - | 61.90 | 73.80 | 61.40 | 78.5 | 51.59 | 10→6 |
| *Open-source models* | | | | | | | | |
| MetaLlama | Llama3.1 70B Instruct | 70 | **83.60** | **95.10** | 80.50 | 75.4 | 59.68 | 6→3 |
| | Llama3.1 70B | 70 | 79.30 | 83.70 | 58.50 | 66.2 | 40.45 | 11→20 |
| | Llama3 70B | 70 | 79.50 | 83.00 | 48.20 | 70.4 | 49.7 | 14→8 |
| | Llama3 70B Instruct | 70 | 82.00 | 93.00 | 81.70 | **82.3** | **61.79** | 4→**2** |
| | CodeLlama 34B Instruct | 34 | - | - | 41.50 | 57 | 39.27 | 17→21 |
| Mistral | Mistral 7B Instruct (v0.3) | 7 | 62.50 | 50.00 | 26.20 | 50.2 | 44.14 | 21→17 |
| | Mixtral 8x7B Instruct | 46.7 | 70.60 | 74.40 | 40.20 | 60.7 | 42.74 | 18 |
| | Codestral 22B | 22 | - | - | 81.10 | 78.2 | 47.61 | 5→13 |
| Phi | Phi3 Medium 128k Instruct | 14 | 78.00 | 91.00 | 62.20 | 75.2 | 48.65 | 9 |
| | Phi3 Mini 128k Instruct | 3.8 | 68.80 | 82.50 | 58.50 | 70 | 39.22 | 11→22 |
| Qwen | Qwen2 7B Instruct | 7 | 70.50 | 82.30 | 79.90 | - | 51.86 | 7→5 |
| | Qwen2 57B-A14B Instruct | 57 | 76.50 | 80.70 | 53.00 | 71.9 | 47.34 | 12→14 |
| | CodeQwen1.5 7B Chat | 7 | - | - | **83.50** | 77.7 | 47.71 | 3→12 |
| Yi | Yi1.5 34B Chat | 34 | 67.62 | 71.70 | 23.20 | 41 | 50.03 | 22→7 |
| | Yi1.5 9B Chat | 9 | 68.40 | 52.30 | 39.00 | 54.4 | 48.15 | 19→10 |
| DeepSeek | DeepSeek Coder 7B Instruct (v1.5) | 7 | 49.20 | 41.00 | 42.10 | 60.7 | 41.59 | 16→19 |
| | DeepSeek Coder 33B Instruct | 33 | - | 60.70 | 79.30 | 70 | 37.45 | 8→23 |
| | DeepSeek Moe 16B Chat | 16.4 | 45.00 | 18.80 | 26.80 | 39.2 | 31.45 | 20→24 |
| | DeepSeek CoderV2 Lite Instruct | 16 | 60.10 | 86.40 | 81.10 | - | 47.12 | 5→15 |
| InternLM | InternLM2.5 20B Chat | 20 | 66.50 | 79.60 | 48.80 | 63 | 46.15 | 13→16 |
| StarCoder | StarCoder2 15B Instruct | 15 | - | - | 46.3 | 66.2 | 47.76 | 15→11 |

## B.1 SELECTION BIAS IN MCQS FORMAT

Building on the findings from Zheng et al. (2024a), which investigated the effects of reordering answer options in multiple-choice questions (MCQs), we observe inconsistent behavior among large language models (LLMs) when performing the same code completion task. Table 8 highlights the sensitivity of LLMs to the order of answers, even for models renowned for their high performance (e.g., GPT, Claude, MetaLlama). Specifically, the results reveal that most models experience significant performance degradation when the correct answer is positioned as "A", with an average performance drop of **25%**. In contrast, placing the correct answer in position "B" leads to a marked performance improvement, with an average increase of **15.49%**.

The standard deviation (STD) further illustrates how differently models respond to answer reordering. For instance, models such as CodeLlama-7B/13B/34B and DeepSeekCoder-33B exhibit substantial dependency on the arrangement of options, whereas models like GPT-4o/3.5, Claude-3, and Claude-3.5 show greater resilience to such selection bias. Interestingly, instruction-tuned models, which are generally expected to demonstrate increased robustness, show minimal to no improvement over their base versions in this regard.

These findings suggest that higher-quality models are more resistant to MCQ biases, reflecting a human-like ability to maintain performance irrespective of answer order. We believe that introducing this MCQ bias into the CodeMMLU benchmark adds an extra layer of difficulty for LLMs, encouraging the research community to prioritize enhancing the consistency and robustness of LLMs.

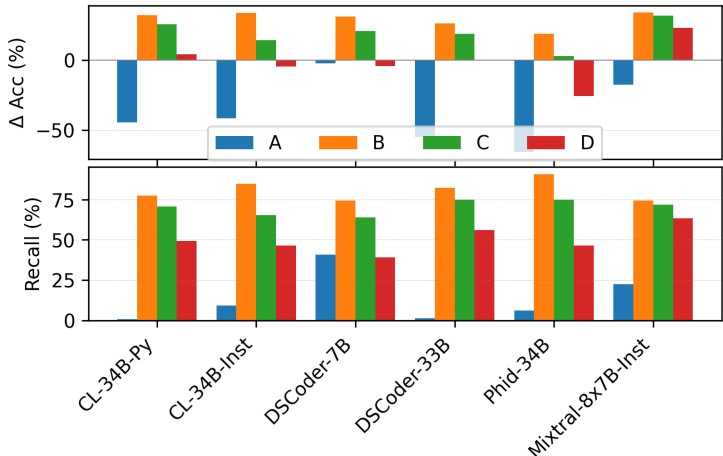

Figure 9: **Task-Specific Accuracy and Performance Fluctuations Across Answer Options** Models exhibit marked fluctuations in accuracy depending on the position of the correct answer in Code Completion in CodeMMLU. Revealing the bias and inconsistencies in related coding multiple-choice question (MCQ) task and how sensitive LLMs are to answer ordering.

Table 8: **Selection bias effect comparison on LLMs.** The performance fluctuation trends show a significant margin of model with high quality and the other. STD stands for standard deviation.

| Models | Instructed | A | B | C | D | STD |
|---|---|---|---|---|---|---|
| GPT-4o | ✓ | 80.49 | 78.05 | 71.34 | 70.12 | 4.38 |
| GPT-3.5-turbo | ✓ | 51.22 | 43.29 | 47.56 | 54.88 | 4.30 |
| Claude3.5 Sonnet | ✓ | 90.24 | 81.1 | 85.37 | 79.27 | **4.23** |
| Claude3.5 Haiku | ✓ | 86.59 | 69.51 | 72.56 | 68.29 | 7.30 |
| Claude3 Opus | ✓ | 79.27 | 77.44 | 82.32 | 84.76 | **2.81** |
| Claude3 Sonnet | ✓ | 62.8 | 64.02 | 73.17 | 73.78 | 5.06 |
| Claude3 Haiku | ✓ | 56.1 | 75 | 73.78 | 76.83 | 8.34 |
| Mixtral 8x7B | ✓ | 22.56 | 74.39 | 71.95 | 63.41 | 20.91 |
| DSCoder 33B | - | 1.22 | 82.32 | 75.00 | 56.10 | 31.75 |
| DSCoder 7B | - | 40.85 | 74.39 | 64.02 | 39.02 | 15.10 |
| Phind-CL 34B | ✓ | 6.10 | 90.85 | 75.00 | 46.34 | 32.21 |
| CL 34B Python | - | 0.61 | 77.44 | 70.73 | 49.39 | 30.09 |
| CL 34B Instruct | ✓ | 9.15 | 84.76 | 65.24 | 46.34 | 27.91 |
| CL 13B Python | - | 0.61 | 54.88 | 70.12 | 12.20 | 28.85 |
| CL 13B Instruct | ✓ | 2.44 | 68.29 | 72.56 | 29.88 | 28.85 |
| CL 7B Python | - | 0.00 | 90.24 | 14.02 | 0.61 | 37.39 |
| CL 7B Instruct | ✓ | 3.66 | 1.22 | 93.90 | 15.85 | 38.07 |

## B.2 CoT might not be all you need

In our experiments with models from over 15 families, we evaluated CodeMMLU under two different prompt settings: standard zero-shot and few-shot, as well as Chain-of-Thought (CoT) with short and long prompts. The detailed results, provided in B.3, reveal a consistent trend of decreased performance in the CoT setting compared to zero-shot and few-shot configurations. Even strong models like GPT-4o and Llama3 70B, known for their robust reasoning and comprehensive capabilities, exhibited this pattern, as illustrated in Figure 13.

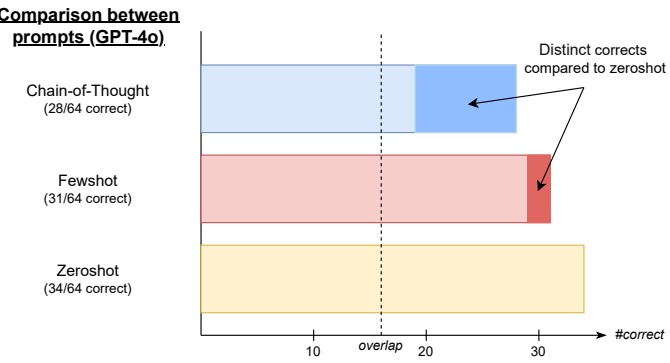

Figure 10: **Comparison experiment of different prompts by GPT-4o in OOP subject.**

A closer analysis of results for zero-shot, few-shot, and CoT prompts, particularly in the GPT-4o experiment on the object-oriented programming (OOP) subject (Figure 10), highlights the effects of an additional "thinking" step introduced by CoT. While this step intended to enhance reasoning, the thought often diverged from the original focus or brought up another challenge for the task. For instance, example 1 in Figure 14 showcases a case where the CoT approach incorporated an unnecessary definitional analysis but failed to arrive at the correct answer. Similarly, examples 2 and 3 (Figure 14) demonstrate misinterpretations during the reasoning process. This additional "thinking" step often introduced new challenges that might lead to incorrect conclusions, resulting in a significant performance gap between zero-shot and CoT prompts.

Interestingly, when visualizing the newly addressed questions by CoT settings (compared to 0-shot), we do not observe a clear pattern of these new corrects. Questions that CoT addressed were neither inherently tricky nor required deeper interpretive reasoning. For example, example 4 (Figure 14) asks for a prediction of program outcomes within the OOP domain, comparable in scope to example 3, yet the CoT approach showed inconsistent performance.

As concluded by the findings in Sprague et al. (2024), the CoT strategy introduces an additional reasoning step that is effective primarily for tasks involving mathematical or logical problem-solving. However, for benchmarks like MMLU, the results with and without CoT prompting are nearly identical. The observed performance decline in the CodeLLM knowledge test set aligns with Sprague et al. (2024), further underscoring the inefficiency of CoT for non-reasoning tasks.

## B.3 Experiment Results

We provide the full experiment results of 43 LLMs from 15 families in Table 9, 10, 11.

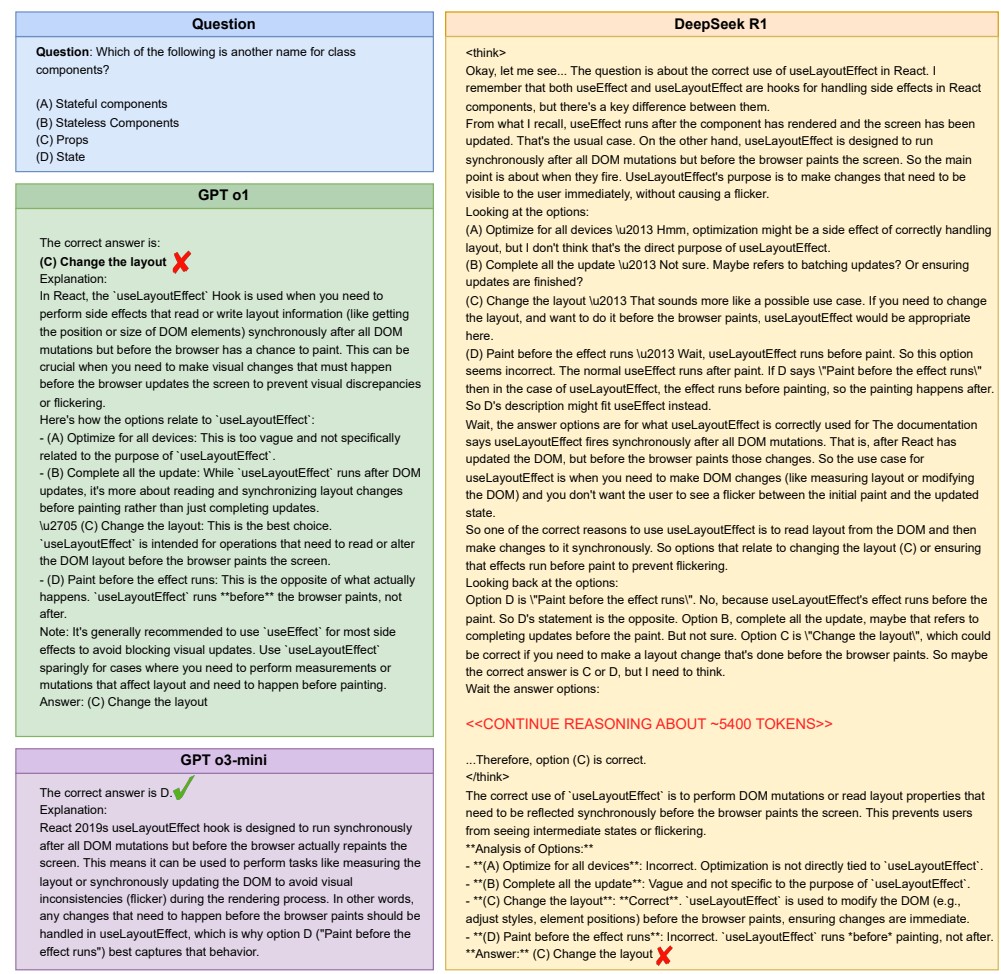

Figure 11: Comparison reasoning process of GPT o1, GPT o3-mini and DeepSeek R1 on knowledge-seeking task.

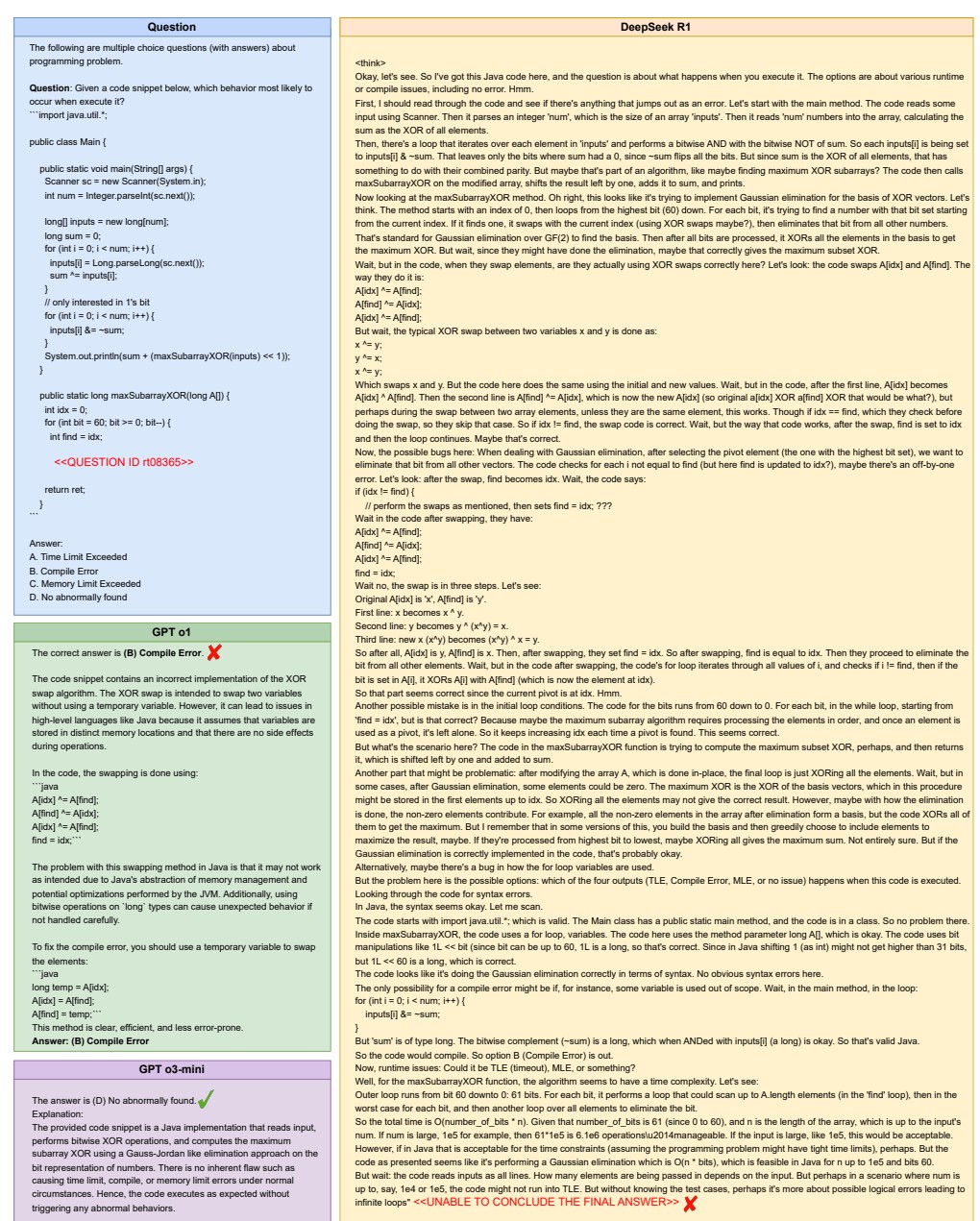

Figure 12: Comparison reasoning process of GPT o1, GPT o3-mini and DeepSeek R1 on problem solving task

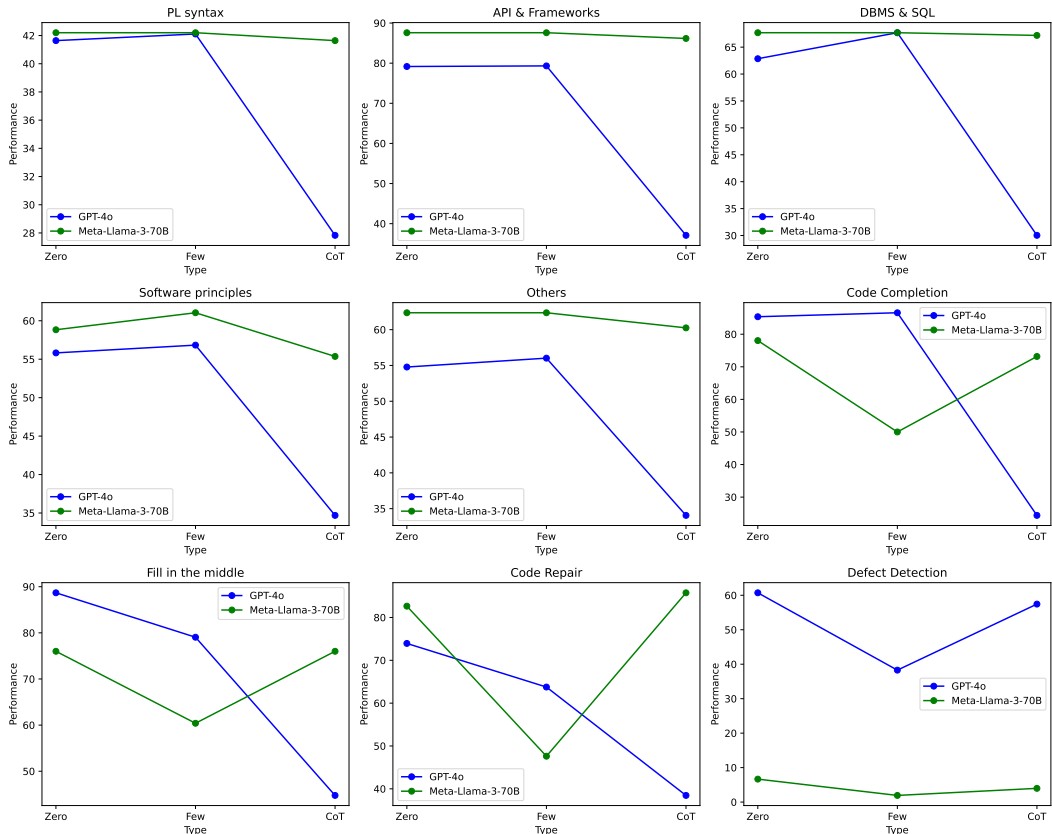

Figure 13: **Comparison between GPT4o and Meta Llama-3 70B on various prompt settings**. We experiment with zero-shot, 1-shot, and CoT prompt configuration, where the result indicates the ineffectiveness of CoT in boosting the models' performance. Comparing to zeroshot config, 1-shot prompt slightly increase the performance in knowledge tasks but falls shorter in real tasks.

**Question**

**Question**: Converting a primitive type data into its corresponding wrapper class object instance is called

(A) Boxing
(B) Wrapping
(C) Instantiation
(D) Autoboxing

**Zeroshot**

(D) Autoboxing ✓

**Chain-of-Though**

To determine the correct answer, let's break down each option and understand what they mean in the context of software development:

(A) Boxing: This term is used to describe the process of converting a primitive type into its corresponding wrapper class object. For example, converting an `int` to an `Integer`.

(B) Wrapping: While this term might sound relevant, it is not commonly used in this specific context. It generally refers to enclosing or encapsulating something within another layer.

(C) Instantiation: This term refers to creating an instance of a class. It does not specifically relate to converting primitive types into their wrapper classes.

(D) Autoboxing: This is a more specific term that describes the automatic conversion that the Java compiler makes between primitive types and their corresponding object wrapper classes. For example, automatically converting an `int` to an `Integer`.

Given these definitions:

- "Boxing" (A) correctly describes manually converting a primitive type into its corresponding wrapper class.
- "Autoboxing" (D), however, specifies that this conversion happens automatically by the compiler.

Since we are looking for just "converting," without specifying whether it's manual or automatic:

Answer:
(A) Boxing ✗

## Question

**Question**: Which among the following is correct for the following code?

```
class A
{
    public : class B
    {
        public : B(int i): data(i)
        {
        }
        int data;
    }
};
class C: public A
{
    class D:public A::B{ };
};
```

(A) Multi-level inheritance is used, with nested classes
(B) Multiple inheritance is used, with nested classes
(C) Single level inheritance is used, with enclosing classes
(D) Single level inheritance is used, with both enclosing and nested classes

## Chain-of-Though

Let's analyze the code step by step to determine which type of inheritance and class structure is being used.

1. **Class A**:
- Contains a nested public class `B`.
- Class `B` has a constructor that initializes an integer member variable `data`.

2. **Class C**:
- Inherits publicly from class `A`.
- Contains a nested public class `D` that inherits publicly from `A::B`.

Now, let's break down the inheritance and nesting:

- **Nested Classes**:
- Class B is nested within Class A.
- Class D is nested within Class C.

- **Inheritance Relationships**:
- Class C inherits from A (single level).
- Nested class D in C inherits from B in A (multi-level).

Given these points, we can evaluate each option:

(A) Multi-level inheritance is used, with nested classes
- This seems correct because there are multiple levels of inheritance involved:
**C -> A** (single level)
**D -> B** (nested multi-level)

(B) Multiple inheritance is used, with nested classes
- This isn't correct because no single class directly inherits from more than one base class.

(C) Single level inheritance is used, with enclosing classes
- This isn't entirely accurate since there are multiple levels of nesting and some multi-level relationships.

(D) Single level inheritance is used, with both enclosing and nested classes
- While it mentions both enclosing and nesting correctly, it incorrectly states "Single level" when there's also multi-level involved due to the relationship between D and B through their respective parent classes.

Therefore,

The correct answer should be:

(A) Multi-level inheritance is used, with nested classes ✗

## Zeroshot

✓ (D) Single level inheritance is used, with both enclosing and nested classes

Explanation:
- The code defines a class `A` which contains a nested class `B`.
- Class `C` inherits from class `A`, making it single-level inheritance.
- Within class `C`, there is another nested class `D` that inherits from the nested class `B` of the base class.

Therefore, this scenario involves single-level inheritance (class C inheriting from A) and includes both enclosing (class A containing B, and C containing D) and nested classes.

## Question

**Question**: What is the output of the following program?

```
class A
{
    protected: int a,b;
    public: void disp()
    {
        cout<<a<<b;
    }
};
class B:public A
{
    int x,y;
};
```

(A) Garbage value
(B) Compile time error
(C) Runtime error
(D) Runs but gives random values as output

## Chain-of-Though

To determine the output of the given program, let's analyze it step by step:
1. **Class Definition**:
   - `class A` has two protected integer members `a` and `b`.
   - It also has a public member function `disp()` that prints the values of `a` and `b`.
2. **Inheritance**:
   - `class B` publicly inherits from class A.
   - Class B introduces two additional private integer members, `x` and `y`.
3. **Functionality**:
   - The function in class A (`disp()`) is designed to print the values of its protected members (`a`, and `b`).
4. **Output Analysis**:
   - Since there are no constructors or any other functions initializing or setting values for variables 'a' and 'b', they will contain whatever data happens to be at their memory locations when an object of class B (or even class A) is created.
5. **Compilation Check**:
   - There are no syntax errors in this code snippet; hence it will compile successfully.
6. **Runtime Behavior**:
   - When you call the disp() method on an instance of either class A or B without initializing 'a' and 'b', they will hold garbage values (uninitialized local variables).

Given these points, we can conclude:
- The program compiles successfully.
- At runtime, since 'a' and 'b' are not initialized before being printed, they will display garbage/random values.

Therefore, considering all steps above:
The correct answer is: (D) Runs but gives random values as output ✗

## Zeroshot

The correct answer is: (B) Compile time error ✓
Explanation: The program will result in a compile-time error because the `cout` statement inside the `disp()` method of class `A` is using the variables `a` and `b`, which are protected members. However, these variables have not been initialized or assigned any values before being used in the output statement. Additionally, there might be an issue with including necessary headers (like `<iostream>`) and using the appropriate namespace (`std::`).

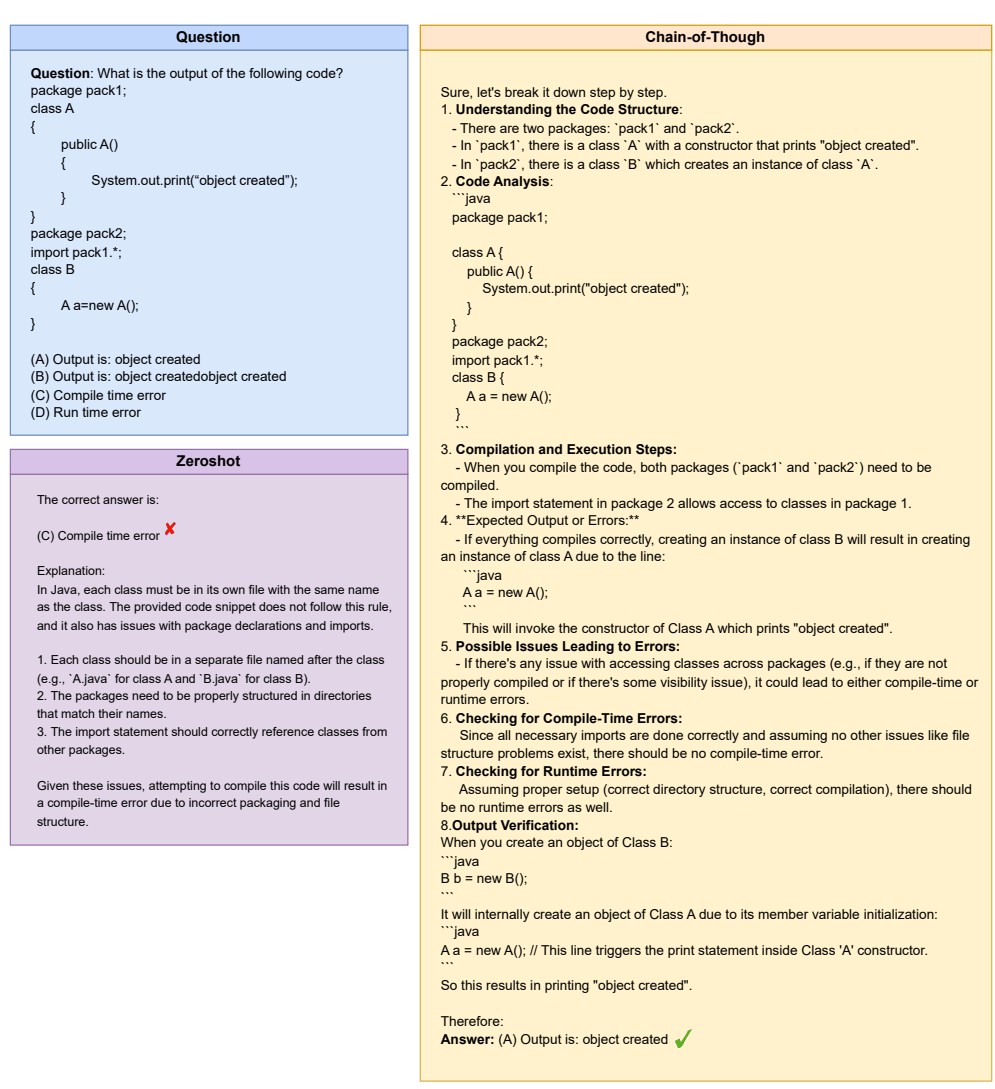

Figure 14: **Examples of Chain-of-Thought and zeroshot in Object-oriented programming (OOP) subset.**

Table 9: **CodeMMLU zeroshot experimental results.**

| Model | API & Frameworks | PL Syntax | Software principle | DBMS & SQL | Others | Code Completion | Fill in the blank | Code Repair | Execution Prediction |
|---|---|---|---|---|---|---|---|---|---|
| Claude3.7 Sonnet | 88.45 | 48.76 | 77.85 | 83.72 | 70.82 | 79.27 | 67.87 | 73.68 | 57.79 |
| Claude3.5 Sonnet | 89.44 | 48.04 | 74.55 | 78.12 | 69.80 | 87.80 | 62.24 | 61.84 | 56.41 |
| Claude3.5 Haiku | 84.59 | 45.26 | 69.23 | 70.99 | 65.28 | 75.00 | 69.94 | 50.00 | 53.16 |
| Claude3 Sonnet | 89.02 | 45.42 | 62.27 | 72.52 | 63.46 | 4.88 | 87.98 | 56.86 | 3.32 |
| GPT o3-mini | 88.45 | 49.10 | 77.46 | 81.68 | 69.66 | 87.20 | 73.27 | 67.11 | 58.33 |
| GPT 4o | 87.02 | 46.08 | 69.93 | 70.99 | 65.21 | 67.68 | 54.39 | 59.21 | 51.27 |
| GPT 4o-mini | 88.45 | 44.18 | 50.12 | 70.23 | 63.82 | 65.85 | 39.50 | 46.05 | 11.97 |
| GPT 3.5-turbo | 84.88 | 38.47 | 51.06 | 58.52 | 51.35 | 76.83 | 29.31 | 38.15 | 36.74 |
| CodeLlama 7B Instruct | 72.04 | 28.23 | 36.80 | 47.84 | 39.02 | 0.00 | 6.95 | 7.90 | 4.26 |
| CodeLlama 7b Python | 54.21 | 23.18 | 35.28 | 43.77 | 31.58 | 42.68 | 17.90 | 10.74 | 6.11 |
| CodeLlama 13B Instruct | 72.04 | 29.12 | 39.09 | 50.89 | 39.02 | 0.00 | 39.97 | 1.35 | 0.79 |
| CodeLlama 13B Python | 0.00 | 25.79 | 14.65 | 0.00 | 0.00 | 51.22 | 70.60 | - | 7.38 |
| CodeLlama 13B | 72.04 | 29.12 | 39.08 | 50.89 | 38.88 | 1.22 | 20.43 | 0.00 | 4.96 |
| CodeLlama 34B Instruct | 79.17 | 34.44 | 41.70 | 52.93 | 46.17 | 37.80 | 24.80 | 27.82 | 3.77 |
| CodeLlama 34B Python | 0.00 | 31.14 | 16.02 | 0.00 | 0.00 | 12.20 | 8.31 | 14.66 | 2.30 |
| Llama3 70B | 86.02 | 40.74 | 52.92 | 63.10 | 56.89 | 21.95 | 92.02 | 24.91 | 2.28 |
| Llama3 70B Instruct | 87.59 | 42.20 | 58.83 | 67.68 | 62.36 | 78.05 | 76.00 | 82.64 | 6.66 |
| Llama3 8B | 75.89 | 32.38 | 42.71 | 53.94 | 46.75 | 48.17 | 94.65 | 66.70 | 5.85 |
| Llama3 8B Instruct | 81.31 | 34.89 | 33.43 | 60.31 | 50.91 | 60.37 | 55.52 | 34.30 | 3.33 |
| Llama3.1 405B Instruct | 91.01 | 46.29 | 72.34 | 74.55 | 68.56 | 88.41 | 54.58 | 59.21 | 57.11 |
| Llama3.1 70B | 87.02 | 41.16 | 53.44 | 65.65 | 57.91 | 3.05 | 4.93 | 22.42 | 2.48 |
| Llama3.1 70B Instruct | 87.30 | 42.40 | 62.71 | 67.18 | 61.20 | 76.83 | 54.49 | 46.05 | 52.11 |
| Llama3.1 8B | 75.04 | 32.86 | 41.94 | 54.71 | 47.63 | 35.37 | 33.26 | 52.60 | 5.11 |
| Llama3.1 8B Instruct | 78.32 | 34.76 | 47.17 | 53.44 | 50.47 | 11.59 | 71.54 | 55.79 | 3.88 |
| Llama3.3 70B Instruct | 87.59 | 39.44 | 50.72 | 62.09 | 54.34 | 81.71 | 63.27 | 38.16 | 18.03 |
| Mistral 7B Instruct (v1) | 74.75 | 30.74 | 41.87 | 53.44 | 43.18 | 59.76 | 79.66 | 23.63 | 2.90 |
| Mistral 7B Instruct (v2) | 71.90 | 32.38 | 42.92 | 54.96 | 46.02 | 43.90 | 10.38 | 47.61 | 2.15 |
| Mistral 7B Instruct (v3) | 75.46 | 33.38 | 47.77 | 57.76 | 48.21 | 56.10 | 16.30 | 52.74 | 2.25 |

| Model | | | | | | | | | |
| --- | --- | --- | --- | --- | --- | --- | --- | --- | --- |
| Mixtral 8x7B Instruct | 85.02 | 37.32 | 49.96 | 61.83 | 52.88 | 3.05 | 25.69 | 33.09 | 4.32 |
| Codestral 22B | 82.17 | 38.50 | 47.62 | 58.52 | 50.18 | 74.39 | | 48.82 | 2.53 |
| Phi-4 | 87.02 | 40.64 | 56.08 | 67.43 | 57.48 | 75.61 | 58.43 | 60.53 | 43.14 |
| Phi-4 Mini Instruct | 87.02 | 36.86 | 48.86 | 61.83 | 54.34 | 23.17 | 23.53 | 35.53 | 18.12 |
| Phi3 Medium Instruct (128k) | 79.89 | 37.19 | 51.44 | 58.78 | 53.46 | 67.68 | 40.82 | 39.57 | 3.48 |
| Phi3 Medium Instruct (4k) | 80.03 | 36.82 | 52.53 | 62.09 | 54.41 | 72.56 | 44.76 | 51.66 | 3.68 |
| Phi3 Mini Instruct (128k) | 74.18 | 31.83 | 44.99 | 54.20 | 46.75 | 42.07 | 31.38 | 14.45 | 1.55 |
| Phi3 Mini Instruct (4k) | 76.32 | 33.14 | 40.70 | 53.94 | 45.30 | 53.66 | 24.00 | 30.53 | 2.35 |
| Phi3 Small Instruct (8k) | 77.89 | 37.32 | 54.40 | 64.12 | 52.22 | 62.80 | 21.42 | 21.07 | 1.95 |
| PhindCL 34B v2 | 79.89 | 35.26 | 44.63 | 51.91 | 45.88 | 53.05 | 27.43 | 18.50 | 3.06 |
| Qwen2.5 14B Instruct | 86.59 | 41.85 | 57.83 | 66.92 | 58.13 | 80.49 | 76.66 | 38.16 | 41.94 |
| QwQ 38B | 78.32 | 39.05 | 52.93 | 59.29 | 51.71 | 35.37 | 40.86 | 36.84 | 42.79 |
| Qwen2 0.5B Instruct | 52.92 | 24.23 | 36.45 | 43.26 | 32.90 | 28.66 | 13.72 | 74.74 | 1.07 |
| Qwen2 1.5B Instruct | 73.04 | 30.03 | 43.52 | 54.20 | 44.78 | 9.76 | 21.28 | 27.20 | 2.49 |
| Qwen2 57B-A14B Instruct | 84.88 | 37.80 | 51.87 | 65.14 | 55.43 | 58.54 | 24.52 | 35.90 | 2.96 |
| Qwen2 7B | 82.88 | 33.73 | 48.12 | 62.85 | 54.70 | 75.00 | 56.93 | 60.64 | 4.63 |
| Qwen2 7B Instruct | 82.74 | 37.06 | 54.41 | 62.85 | 53.98 | 69.51 | 50.78 | 46.05 | 4.31 |
| QwenCoder2.5 32B Instruct | 86.88 | 46.54 | 71.15 | 73.28 | 65.35 | 81.71 | 68.58 | 38.16 | 48.12 |
| QwenCoder2.5 14B Instruct | 87.45 | 41.58 | 58.08 | 63.10 | 55.51 | 82.93 | 73.41 | 55.26 | 31.35 |
| CodeQwen1.5 7B | 74.47 | 30.56 | 38.89 | 51.65 | 40.41 | 39.63 | 5.40 | - | 5.92 |
| CodeQwen1.5 7B Chat | 67.62 | 31.70 | 38.25 | 46.06 | 39.24 | 89.02 | 37.06 | - | 3.38 |
| Yi1.5 34B Chat | 79.60 | 37.03 | 52.93 | 61.32 | 52.52 | 67.68 | 35.09 | 55.30 | 3.00 |
| Yi-1.5 6B Chat | 75.75 | 34.45 | 45.19 | 58.02 | 49.53 | 53.05 | 41.90 | 36.66 | 2.67 |
| Yi1.5 9B Chat | 75.46 | 35.82 | 52.32 | 60.05 | 52.81 | 60.98 | 32.27 | 49.90 | 5.43 |
| DeepSeek-R1 | 71.47 | 39.12 | 56.12 | 63.87 | 56.09 | 56.71 | 33.26 | 42.11 | 39.23 |
| DeepSeek-V3 | 87.02 | 43.94 | 54.85 | 66.92 | 60.39 | 56.10 | 41.33 | 51.32 | 46.00 |
| DSCoder 33B | 0.00 | 0.00 | 15.98 | 0.00 | 0.00 | 6.10 | 13.62 | 18.64 | 5.83 |
| DSCoder 33B Instruct | 74.89 | 32.40 | 41.24 | 52.16 | 42.89 | 8.54 | 31.94 | 40.06 | 5.32 |
| DSCoder 6.7B | 70.19 | 28.71 | 36.96 | 49.36 | 39.10 | 16.46 | 0.09 | 2.63 | 0.02 |
| DSCoder 6.7B Instruct | 71.75 | 29.84 | 38.01 | 52.67 | 38.15 | 26.83 | 3.66 | 0.00 | 3.69 |
| DSCoder 7B (v1.5) | 81.60 | 35.98 | 47.21 | 57.51 | 46.32 | 0.00 | 27.62 | 41.13 | 0.00 |
| DSCoder 7B Instruct (v1.5) | 79.74 | 33.60 | 43.29 | 55.47 | 44.93 | 13.41 | 31.09 | 65.49 | 3.85 |

| | | | | | | | | | |
|---|---|---|---|---|---|---|---|---|---|
| DSMoE 16B | 56.49 | 23.47 | 31.81 | 43.00 | 34.87 | 0.61 | 26.21 | 46.05 | 1.25 |
| DSMoE 16B Chat | 41.94 | 21.54 | 33.44 | 41.48 | 31.36 | 18.90 | 50.54 | 37.21 | 2.65 |
| DSCoderV2 Lite | 83.88 | 35.00 | 47.34 | 59.29 | 48.50 | 40.85 | 19.96 | 32.88 | 0.17 |
| DSCoderV2 Lite Instruct | 81.74 | 38.07 | 49.85 | 64.38 | 50.04 | 46.34 | 45.04 | 39.57 | 3.53 |
| InternLM2.5 20B Chat | 78.03 | 37.67 | 51.85 | 60.05 | 54.63 | 57.93 | 24.47 | 38.22 | 1.12 |
| InternLM2.5 7B Chat | 79.32 | 35.32 | 50.93 | 56.74 | 51.71 | 59.15 | 18.69 | 26.47 | 5.41 |
| StarCoder2 15B Instruct | 78.74 | 34.41 | 46.94 | 52.42 | 47.85 | 73.17 | 42.04 | 52.53 | 3.40 |
| StarCoder2 7B | 63.77 | 27.97 | 34.19 | 48.85 | 36.25 | 30.49 | 67.78 | 6.62 | 4.81 |

Table 10: **CodeMMLU few-shot experimental results.**

| Model | PL syntax | API & Frameworks | DBMS & SQL | Software principles | Others | Code Completion | Fill in the blank | Code Repair | Execution Prediction |
|---|---|---|---|---|---|---|---|---|---|
| Claude3 Sonnet | 44.58 | 86.31 | 70.99 | 64.73 | 63.68 | 11.59 | 40.11 | 52.67 | 3.11 |
| GPT-4o | 42.11 | 79.32 | 67.68 | 56.83 | 56.02 | 86.59 | 79.05 | 63.76 | 38.28 |
| GPT-3.5-turbo | 38.47 | 84.88 | 58.52 | 52.93 | 50.04 | 70.12 | 54.06 | 35.38 | 34.45 |
| CodeLlama 13B Instruct | 29.12 | 72.04 | 50.89 | 39.09 | 39.02 | 62.80 | 17.61 | 25.71 | 3.03 |
| CodeLlama 13B Python | 25.79 | 62.62 | 44.02 | 35.16 | 33.41 | 0.61 | 0.00 | 1.28 | 0.03 |
| CodeLlama 13B | 29.12 | 72.04 | 50.89 | 42.04 | 38.88 | 0.61 | 0.19 | 0.00 | 0.00 |
| CodeLlama 34B Instruct | 34.44 | 79.17 | 52.93 | 44.51 | 46.17 | 10.37 | 0.00 | 27.03 | 0.07 |
| CodeLlama 34B Python | 31.14 | 74.18 | 50.13 | 39.94 | 42.52 | 1.22 | 0.00 | 1.28 | 0.69 |
| CodeLlama 7B Instruct | 28.23 | 72.04 | 47.84 | 39.56 | 39.02 | 1.22 | 44.20 | 6.69 | 0.02 |
| CodeLlama 7B Python | 23.49 | 54.07 | 44.27 | 37.23 | 31.80 | 0.00 | 0.00 | 0.00 | 0.12 |
| Llama3 70B | 40.74 | 86.02 | 63.10 | 55.47 | 56.89 | 67.07 | 25.50 | 12.02 | 0.80 |
| Llama3 70B Instruct | 42.20 | 87.59 | 67.68 | 61.04 | 62.36 | 50.00 | 60.40 | 47.61 | 1.93 |
| Llama3 8B | 32.38 | 75.89 | 53.94 | 45.52 | 46.75 | 93.90 | 40.44 | 92.31 | 3.62 |
| Llama3 8B Instruct | 34.77 | 81.74 | 60.81 | 50.30 | 50.62 | 27.44 | 87.69 | 17.36 | 0.28 |
| Llama3.1 70B | 41.16 | 87.02 | 65.65 | 56.49 | 57.91 | 48.78 | 2.07 | 18.16 | 0.75 |
| Llama3.1 70B Instruct | 41.38 | 87.45 | 69.21 | 60.98 | 59.30 | 6.10 | 30.39 | 80.15 | 4.31 |
| Llama3.1 8B | 32.86 | 75.04 | 54.71 | 45.13 | 47.63 | 1.22 | 0.28 | 2.63 | 0.03 |
| Llama3.1 8B Instruct | 34.76 | 78.32 | 53.44 | 48.73 | 50.47 | 50.00 | 67.36 | 57.52 | 3.53 |
| Mistral 7B Instruct (v1) | 30.74 | 74.75 | 53.44 | 44.76 | 43.18 | 14.02 | 7.84 | 44.53 | 2.13 |
| Mistral 7B Instruct (v2) | 32.38 | 71.90 | 54.96 | 45.93 | 46.02 | 23.78 | 5.31 | 7.97 | 0.75 |
| Mistral 7B Instruct (v3) | 33.38 | 75.46 | 57.76 | 50.27 | 48.21 | 19.51 | 2.35 | 9.32 | 0.84 |
| Mixtral 8x7B Instruct | 37.33 | 84.74 | 61.07 | 53.24 | 53.32 | 25.00 | 7.23 | 9.32 | 1.83 |
| Codestral 22B | 38.50 | 82.17 | 58.52 | 50.35 | 50.18 | 31.71 | 14.61 | 21.00 | 0.97 |
| Phi3 Medium Instruct (128k) | 37.19 | 79.89 | 58.78 | 53.27 | 53.46 | 10.37 | 4.04 | 2.70 | 0.22 |
| Phi3 Medium Instruct (4k) | 36.82 | 80.03 | 62.09 | 54.92 | 54.41 | 7.32 | 6.34 | 29.66 | 0.50 |
| Phi3 Mini Instruct (128k) | 31.83 | 74.18 | 54.20 | 47.29 | 46.75 | 1.22 | 0.38 | 2.56 | 0.07 |
| Phi3 Mini Instruct (4k) | 33.14 | 76.32 | 53.94 | 44.01 | 45.30 | 0.00 | 0.19 | 0.00 | 0.05 |

| | 37.32 | 78.60 | 64.12 | 56.83 | 52.22 | 56.10 | 56.65 | 38.77 | 1.90 |
|---|---|---|---|---|---|---|---|---|---|
| Phi3 Small Instruct (8k) | | | | | | | | | |
| Qwen2 57B-A14B Instruct | 37.80 | 84.88 | 65.14 | 55.19 | 55.43 | 42.07 | 19.26 | 14.59 | 1.14 |
| Qwen2 7B | 36.38 | 82.88 | 61.58 | 53.06 | 54.49 | 7.32 | 2.54 | 1.28 | 0.00 |
| Qwen2 7B Instruct | 37.01 | 82.74 | 62.85 | 56.35 | 54.27 | 35.98 | 43.68 | 10.60 | 2.91 |
| CodeQwen1.5 7B | 30.56 | 74.04 | 51.15 | 42.46 | 40.34 | 0.00 | 0.14 | 0.00 | 0.07 |
| CodeQwen1.5 7B Chat | 31.70 | 67.62 | 46.06 | 40.17 | 39.10 | 1.83 | 3.05 | 8.04 | 1.58 |
| Yi1.5 34B Chat | 37.03 | 80.17 | 62.34 | 54.93 | 52.81 | 21.95 | 13.01 | 21.28 | 0.25 |
| Yi-1.5 6B Chat | 34.34 | 75.75 | 58.02 | 48.51 | 49.53 | 0.61 | 5.26 | 15.73 | 1.13 |
| Yi1.5 9B Chat | 35.74 | 75.46 | 60.05 | 54.19 | 52.81 | 0.61 | 24.99 | 3.98 | 0.07 |
| DSCoder 33B | 29.26 | 74.75 | 48.60 | 40.93 | 40.41 | 0.61 | 0.00 | 1.28 | 0.00 |
| DSCoder 33B Instruct | 32.35 | 75.32 | 51.91 | 43.73 | 43.03 | 14.63 | 9.02 | 2.56 | 1.22 |
| DSCoder 6.7B | 28.71 | 70.19 | 49.36 | 40.06 | 39.10 | 1.22 | 0.28 | 1.28 | 0.00 |
| DSCoder 6.7B Instruct | 29.84 | 71.75 | 52.67 | 41.68 | 38.15 | 5.49 | 0.89 | 1.28 | 0.02 |
| DSCoder 7B (v1.5) | 36.08 | 81.31 | 57.51 | 49.79 | 46.46 | 37.80 | 63.74 | 23.22 | 2.60 |
| DSCoder 7B Instruct (v1.5) | 33.46 | 79.60 | 54.71 | 45.86 | 44.93 | 0.00 | 0.00 | 0.00 | 0.02 |
| DSMoE 16B | 23.89 | 56.49 | 43.51 | 35.59 | 34.72 | 0.00 | 0.09 | 0.00 | 0.00 |
| DSMoE 16B Chat | 21.54 | 43.37 | 41.73 | 35.87 | 31.15 | 0.61 | 0.05 | 0.00 | 0.08 |
| DSCoderV2 Lite | 35.00 | 83.88 | 59.29 | 50.33 | 48.50 | 0.61 | 0.00 | 1.28 | 0.00 |
| DSCoderV2 Lite Instruct | 38.07 | 81.74 | 64.38 | 53.48 | 50.04 | 7.93 | 14.09 | 2.63 | 0.17 |
| InternLM2.5 20B Chat | 37.67 | 78.03 | 60.05 | 53.90 | 54.63 | 25.61 | 1.03 | 1.35 | 0.02 |
| StarCoder2 15B Instruct | 34.41 | 78.89 | 52.93 | 47.85 | 47.70 | 4.27 | 4.23 | 2.63 | 0.17 |

Table 11: **CodeMMLU Chain-of-Though with zeroshot experimental results.**

| Model | PL syntax | API & Frameworks | DBMS & SQL | Software principles | Others | Code Completion | Fill in the middle | Code Repair | Execution Prediction |
|---|---|---|---|---|---|---|---|---|---|
| Claude3 Sonnet | 42.60 | 41.94 | 71.76 | 72.81 | 61.20 | 0.61 | 61.34 | 48.61 | 3.26 |
| GPT-4o | 27.83 | 37.09 | 30.03 | 34.69 | 34.06 | 24.39 | 44.76 | 38.46 | 57.44 |
| GPT-3.5-turbo | 0.79 | 85.73 | 60.81 | 49.00 | 50.47 | 69.51 | 39.36 | 48.82 | 41.95 |
| CodeLlama 7B Instruct | 27.30 | 69.47 | 45.55 | 38.43 | 37.93 | 37.20 | 35.23 | 29.18 | 2.70 |
| CodeLlama 7B Python | 24.49 | 59.49 | 42.24 | 30.03 | 30.49 | 56.71 | 0.00 | 51.35 | 0.00 |
| CodeLlama 13B Instruct | 30.06 | 76.89 | 50.64 | 37.65 | 40.04 | 20.12 | 40.02 | 65.70 | 2.45 |
| CodeLlama 13B Python | 27.17 | 69.33 | 46.06 | 30.03 | 36.11 | 0.00 | 0.00 | 100.00 | 0.00 |
| CodeLlama 13B | 30.05 | 76.75 | 50.89 | 37.60 | 40.04 | 0.61 | 0.52 | 3.92 | 0.03 |
| CodeLlama 34B Instruct | 34.00 | 77.32 | 54.45 | 42.76 | 45.37 | 79.27 | 41.94 | 18.16 | 2.32 |
| CodeLlama 34B Python | 23.87 | 74.75 | 51.91 | 32.06 | 40.34 | 0.00 | 1.69 | 30.46 | 0.77 |
| Llama3 70B | 30.80 | 72.90 | 50.64 | 38.03 | 40.70 | 39.63 | 19.16 | 0.00 | 0.53 |
| Llama3 70B Instruct | 30.85 | 68.19 | 45.04 | 38.12 | 36.11 | 89.63 | 46.97 | 98.72 | 2.84 |
| Llama3 8B | 39.84 | 85.73 | 63.10 | 52.23 | 55.73 | 19.51 | 92.02 | 100.00 | 3.53 |
| Llama3 8B Instruct | 41.64 | 86.16 | 67.18 | 55.36 | 60.25 | 73.17 | 76.00 | 85.76 | 3.98 |
| Llama3.1 70B | 31.48 | 76.60 | 56.74 | 41.05 | 46.32 | 11.59 | 94.65 | 79.87 | 2.65 |
| Llama3.1 70B Instruct | 33.54 | 78.60 | 55.98 | 46.06 | 49.02 | 59.76 | 13.81 | 33.02 | 2.15 |
| Llama3.1 8B | 40.50 | 86.16 | 64.12 | 52.21 | 55.87 | 19.51 | 4.93 | 45.15 | 1.62 |
| Llama3.1 8B Instruct | 40.10 | 78.89 | 65.14 | 53.50 | 57.55 | 85.98 | 84.64 | 82.99 | 3.70 |
| Mistral 7B Instruct (v1) | 31.98 | 75.04 | 55.98 | 43.99 | 46.24 | 23.17 | 17.57 | 38.15 | 0.00 |
| Mistral 7B Instruct (v2) | 33.41 | 78.74 | 54.20 | 46.02 | 46.54 | 72.56 | 71.54 | 44.70 | 2.81 |
| Mistral 7B Instruct (v3) | 26.53 | 63.91 | 50.64 | 35.32 | 33.84 | 75.00 | 62.61 | 74.95 | 4.53 |
| Mixtral 8x7B Instruct | 31.45 | 69.90 | 51.65 | 41.07 | 43.84 | 37.80 | 23.44 | 47.68 | 1.92 |
| Codestral 22B | 32.11 | 70.47 | 52.67 | 39.97 | 44.27 | 57.32 | 26.30 | 55.37 | 2.13 |
| Phi3 Medium Instruct (128k) | 35.16 | 79.60 | 55.73 | 46.01 | 47.78 | 33.54 | 60.40 | 81.70 | 3.21 |
| Phi3 Medium Instruct (4k) | 38.22 | 79.32 | 58.52 | 49.15 | 48.94 | 80.49 | 46.17 | 60.78 | 2.45 |
| Phi3 Mini Instruct (128k) | 37.30 | 79.32 | 57.76 | 50.21 | 51.79 | 85.98 | 47.11 | 73.87 | 3.28 |
| Phi3 Mini Instruct (4k) | 35.69 | 78.03 | 57.76 | 47.43 | 49.60 | 85.37 | 45.89 | 60.64 | 2.46 |

| | | | | | | | | |
|---|---|---|---|---|---|---|---|---|
| Phi3 Small Instruct (8k) | 30.21 | 65.05 | 50.64 | 40.43 | 40.92 | 67.07 | 30.62 | 60.57 | 2.97 |
| PhindCL 34B v2 | 30.74 | 62.77 | 47.33 | 37.82 | 40.04 | 57.93 | 29.92 | 31.60 | 2.50 |
| Qwen2 0.5B Instruct | 36.74 | 78.60 | 64.89 | 56.41 | 53.10 | 43.29 | 41.99 | 43.83 | 3.03 |
| Qwen2 1.5B Instruct | 35.23 | 80.60 | 55.22 | 45.82 | 45.51 | 0.61 | 36.59 | 27.89 | 1.42 |
| Qwen2 57B-A14B Instruct | 22.14 | 48.50 | 40.97 | 31.37 | 29.61 | 37.20 | 13.72 | 21.28 | 1.27 |
| Qwen2 7B | 28.15 | 68.76 | 53.69 | 41.37 | 41.50 | 27.44 | 21.28 | 36.66 | 3.53 |
| Qwen2 7B Instruct | 31.93 | 72.33 | 56.74 | 41.35 | 45.15 | 72.56 | 40.16 | 57.73 | 3.70 |
| CodeQwen1.5 7B | 36.96 | 82.88 | 64.63 | 51.45 | 54.12 | 75.00 | 77.64 | 58.14 | 4.18 |
| CodeQwen1.5 7B Chat | 32.32 | 77.60 | 57.25 | 47.22 | 48.29 | 52.44 | 76.00 | 60.64 | 5.31 |
| Yi1.5 34B Chat | 35.69 | 77.46 | 54.45 | 45.61 | 47.92 | 87.20 | 76.33 | 68.75 | 4.66 |
| Yi-1.5 6B Chat | 32.19 | 70.04 | 58.52 | 42.17 | 45.08 | 65.85 | 68.11 | 28.76 | 3.25 |
| Yi1.5 9B Chat | 34.45 | 73.47 | 56.49 | 52.42 | 49.45 | 71.95 | 75.95 | 19.58 | 3.08 |
| DSCoder 6.7B | 26.51 | 60.49 | 44.78 | 30.98 | 31.58 | 0.00 | 0.05 | 0.00 | 0.00 |
| DSCoder 6.7B Instruct | 29.44 | 68.90 | 49.11 | 34.37 | 36.47 | 85.37 | 47.39 | 77.51 | 3.10 |
| DSCoder 7B (v1.5) | 35.24 | 79.32 | 54.96 | 44.78 | 42.30 | 0.00 | 27.38 | 55.72 | 2.08 |
| DSCoder 7B Instruct (v1.5) | 33.22 | 79.32 | 52.42 | 43.82 | 45.08 | 50.00 | 45.66 | 29.11 | 3.29 |
| DSCoder 33B | 28.50 | 71.33 | 47.84 | 38.36 | 38.07 | 0.00 | 0.00 | 3.92 | 0.44 |
| DSCoder 33B Instruct | 31.54 | 75.75 | 49.11 | 38.84 | 41.58 | 45.12 | 28.51 | 31.46 | 2.48 |
| DSMoE 16B | 21.25 | 49.50 | 38.93 | 33.48 | 29.98 | 0.00 | 0.00 | 0.00 | 0.13 |
| DSMoE 16B Chat | 23.81 | 60.77 | 48.09 | 37.37 | 36.83 | 29.27 | 36.17 | 76.65 | 2.20 |
| DSCoderV2 Lite | 33.78 | 83.59 | 57.51 | 47.19 | 46.10 | 29.88 | 0.80 | 64.62 | 0.00 |
| DSCoderV2 Lite Instruct | 22.22 | 38.66 | 28.24 | 25.64 | 20.50 | 91.46 | 55.47 | 62.99 | 2.63 |
| InternLM2.5 20B Chat | 35.00 | 75.75 | 57.76 | 45.70 | 50.04 | 81.10 | 75.90 | 64.48 | 3.58 |
| InternLM2.5 7B Chat | 31.30 | 67.76 | 50.38 | 42.03 | 42.01 | 79.27 | 49.98 | 60.64 | 3.75 |
| StarCoder2 15B Instruct | 34.44 | 83.31 | 56.23 | 46.99 | 46.61 | 75.00 | 53.31 | 24.91 | 2.30 |
| StarCoder2 7B | 27.77 | 63.77 | 47.33 | 36.95 | 34.72 | 1.22 | 0.05 | 1.35 | 0.08 |

### B.4 CODEMMLU EXAMPLE

**General knowledge MCQ example:**

> The following are multiple-choice questions (with answers) about debugging a programming problem.
> **Question:** Suppose we have an O(n) time algorithm that finds the median of an unsorted array. Now consider a QuickSort implementation where we first find the median using the above algorithm, then use the median as a pivot. What will be the worst-case time complexity of this modified QuickSort?
> (A) $O(n^2 \log n)$
> (B) $O(n^2)$
> (C) $O(n \log n \log n)$
> (D) $O(n \log n)$

**Code Completion example:**

> The following are multiple-choice questions (with answers) about programming problems.
> **Question:** Which solution below is the most likely to complete the following code to achieve the desired goal?
>
> ```python
> from typing import List
>
> def has_close_elements(numbers: List[float], threshold: float) -> bool:
>     """ Check if in given list of numbers, are any two numbers closer to each
> ↪ other than given threshold.
>     >>> has_close_elements([1.0, 2.0, 3.0], 0.5)
>     False
>     >>> has_close_elements([1.0, 2.8, 3.0, 4.0, 5.0, 2.0], 0.3)
>     True
>     """
> ```
>
> (A)
> ```python
>     for i in range(len(numbers)):  # Change range to len(numbers)
>         for j in range(i + 1, len(numbers)):
>             if abs(numbers[i] - numbers[j]) < threshold:
>                 return True
>         return False
> ```
>
> (B)
> ```python
>     return any(abs(a - b) < threshold for a, b \
>         in zip(numbers, numbers[1:]))
> ```
>
> (C)
> ```python
>     for i in range(len(numbers) - 1):
>         for j in range(i + 1, len(numbers)):
>             if abs(numbers[i] - numbers[j]) > threshold:
>                 return False
>         return True
> ```
>
> (D)
> ```python
>     for idx, elem in enumerate(numbers):
>         for idx2, elem2 in enumerate(numbers):
>             if idx != idx2:
>                 distance = abs(elem - elem2)
>                 if distance < threshold:
>                     return True
>
>     return False
> ```
>
> **Answer**:

**Fill in the blank example:**

The following are multiple-choice questions (with answers) about a programming problem with incomplete solution.

**Problem statement:** You are given an array of intervals, where intervals[i] = [starti, endi] and each starti is unique. The right interval for an interval i is an interval j such that startj >= endi and startj is minimized. Note that i may equal j. Return an array of right interval indices for each interval i. If no right interval exists for interval i, then put -1 at index i.

**Incomplete Solution:**

```python
def find_right_interval(intervals):
    n = len(intervals)
    res = [-1] * n
    for i in range(n):
        intervals[i].append(i)

    def binary_search(ele):
        left, right = 0, n-1
        ans = float('inf')
        while left <= right:
            mid = (left + right) // 2
            if intervals[mid][0] >= ele:
                ans = min(ans, mid)
                right = mid - 1
            else:
                left = mid + 1
        return ans

    intervals.sort()
    for i in intervals:
        -----------------
        if val != float('inf'):
            res[i[2]] = intervals[val][2]
    return res
```

**Question:** The provided solution is missing a part, which option below is the most likely to complete the solution and achieve the desired goal?

(A)
```python
    val = binary_search(i[1])
```

(B)
```python
    if val != float('inf'):
```

(C)
```python
    val = binary_search(i[1])
```

(D)
```python
    if val != float('inf'):
      res[i[2]] = intervals[val][2]
```

**Answer**:

**Code Repair example:**

The following are multiple-choice questions (with answers) about debugging a programming problem.

**Question:** The following code snippet is producing incorrect results; Which solution below correctly identifies the bug and repairs it to achieve the desired goal?

```java
1  import java.util.*;
2  public class DETECT_CYCLE {
3      public static boolean detect_cycle(Node node) {
4          Node hare = node;
5          Node tortoise = node;
6          while (true) {
7              if (hare.getSuccessor() == null)
8                  return false;
9              tortoise = tortoise.getSuccessor();
10             hare = hare.getSuccessor().getSuccessor();
11             if (hare == tortoise)
12                 return true;
```

```
13        }
14      }
15 }
```

(A) Modify line 6:

```
    for (; ; ) {
```

(B) Modify line 7:

```
        if (null==hare ||hare.getSuccessor() == null)
```

(C) Modify line 12:

```
    return hare.getSuccessor() != null && hare == tortoise;
```

(D) Modify line 11:

```
        if (Objects.equals(hare, tortoise))
```

**Execution Prediction example:**

The following are multiple-choice questions (with answers) about programming problem.
**Question:** Given a code snippet below, which behavior most likely to occur when running the solution?

```
import java.util.*;
public class Main {
    public static void main(string[] args) {
        Scanner sc = new Scanner(System.in);
        int A = sc.nextInt();
        int B = sc.nextInt();
        int T = sc.nextInt();
        int S = T/A System.out.println(s*b);
    }
}
```

(A) Memory Limit Exceeded
(B) Runtime Error
(C) Compile Error
(D) No abnormally found

## C   MODELS SETUP

In our experiment and study, we consider GPT-4o (OpenAI et al., 2024), GPT-3.5 (OpenAI, 2023), Claude-3.5, Claude-3 (Anthropic, 2024), MetaLlama 3.1 (Dubey et al., 2024), MetaLlama3 (Dubey et al., 2024), CodeLLaMA (Rozière et al., 2024), DeepSeek AI, DeepSeek Coder, DeepSeek Coder V2 (Guo et al., 2024b; DeepSeek-AI et al., 2024; Guo et al., 2024a), MistralAI, Codetral (Jiang et al., 2024), Qwen2 (Yang et al., 2024), CodeQwen1.5 (Bai et al., 2023), Yi (AI et al., 2024), StarCoder2 (Lozhkov et al., 2024a), InternLM (Cai et al., 2024), Phind (Phind, 2023).

Table 12: Language Models Description

|  | Model ID | Short Name | Link |
|---|---|---|---|
| OpenAI | GPT-o1-2024-12-17 | GPT o1 | - |
|  | GPT-o3-mini-2025-01-31 | GPT o3-mini | - |
|  | GPT-4o-2024-05-13 | GPT 4o | - |
|  | GPT-4o-mini-2024-07-18 | GPT 4o mini | - |
|  | GPT-3.5-turbo-16k-0613 | GPT-3.5-turbo | - |
| Anthropic | Claude-3.5-sonnet-20241022 | Claude3.5 Sonnet | - |
|  | Claude-3.7-sonnet-20250219 | Claude3.7 Sonnet | - |
|  | Claude-3.5-haiku-20241022 | Claude3.5 Haiku | - |
|  | Claude-3-haiku-20240307 | Claude3 Haiku | - |

| | | | |
|---|---|---|---|
| | Claude-3-sonnet-202402029 | Claude3 Sonnet | - |
| | Claude-3-opus-20240229 | Claude3 Opus | - |
| CodeLlama | codellama/CodeLlama-13b-Instruct-hf | CodeLlama 13B Instruct | codellama/CodeLlama-13b-Instruct-hf |
| | codellama/CodeLlama-13b-Python-hf | CodeLlama 13B Python | codellama/CodeLlama-13b-Python-hf |
| | codellama/CodeLlama-13b-hf | CodeLlama 13B | codellama/CodeLlama-13b-hf |
| | codellama/CodeLlama-34b-Instruct-hf | CodeLlama 34B Instruct | codellama/CodeLlama-34b-Instruct-hf |
| | codellama/CodeLlama-34b-Python-hf | CodeLlama 34B Python | codellama/CodeLlama-34b-Python-hf |
| | codellama/CodeLlama-7b-Instruct-hf | CodeLlama 7B Instruct | codellama/CodeLlama-7b-Instruct-hf |
| | codellama/CodeLlama-7b-Python-hf | CodeLlama 7B Python | codellama/CodeLlama-7b-Python-hf |
| MetaLlama | meta-llama/Meta-Llama-3-70B | Llama3 70B | meta-llama/Meta-Llama-3-70B |
| | meta-llama/Meta-Llama-3-70B-Instruct | Llama3 70B Instruct | meta-llama/Meta-Llama-3-70B-Instruct |
| | meta-llama/Meta-Llama-3-8B | Llama3 8B | meta-llama/Meta-Llama-3-8B |
| | meta-llama/Meta-Llama-3-8B-Instruct | Llama3 8B Instruct | meta-llama/Meta-Llama-3-8B-Instruct |
| | meta-llama/Meta-Llama-3.1-70B | Llama3.1 70B | meta-llama/Meta-Llama-3.1-70B |
| | meta-llama/Meta-Llama-3.1-70B-Instruct | Llama3.1 70B Instruct | meta-llama/Meta-Llama-3.1-70B-Instruct |
| | meta-llama/Meta-Llama-3.1-8B | Llama3.1 8B | meta-llama/Meta-Llama-3.1-8B |
| | meta-llama/Meta-Llama-3.1-8B-Instruct | Llama3.1 8B Instruct | meta-llama/Meta-Llama-3.1-8B-Instruct |
| | meta-llama/Meta-Llama-3.1-405B-Instruct | Llama3.1 405B Instruct | meta-llama/Meta-Llama-3.1-405B-Instruct |
| | meta-llama/Meta-Llama-3.3-70B-Instruct | Llama3.3 70B Instruct | meta-llama/Meta-Llama-3.3-70B-Instruct |
| Mistral | mistralai/Mistral-7B-Instruct-v0.1 | Mistral 7B Instruct (v1) | mistralai/Mistral-7B-Instruct-v0.1 |
| | mistralai/Mistral-7B-Instruct-v0.2 | Mistral 7B Instruct (v2) | mistralai/Mistral-7B-Instruct-v0.2 |
| | mistralai/Mistral-7B-Instruct-v0.3 | Mistral 7B Instruct (v3) | mistralai/Mistral-7B-Instruct-v0.3 |
| | mistralai/Mixtral-8x7B-Instruct-v0.1 | Mixtral 8x7B Instruct | mistralai/Mixtral-8x7B-Instruct-v0.1 |
| | mistralai/Codestral-22B-v0.1 | Codestral 22B | mistralai/Codestral-22B-v0.1 |
| Phi | microsoft/phi-4 | Phi-4 | microsoft/phi-4 |
| | microsoft/Phi-4-mini-instruct | Phi-4-mini-instruct | microsoft/Phi-4-mini-instruct |
| | microsoft/Phi-3-medium-128k-instruct | Phi3 Medium Instruct (128k) | microsoft/Phi-3-medium-128k-instruct |
| | microsoft/Phi-3-medium-4k-instruct | Phi3 Medium Instruct (4k) | microsoft/Phi-3-medium-4k-instruct |

| | microsoft/Phi-3-mini-128k-instruct
microsoft/Phi-3-mini-4k-instruct
microsoft/Phi-3-small-8k-instruct | Phi3 Mini Instruct (128k)
Phi3 Mini Instruct (4k)
Phi3 Small Instruct (8k) | microsoft/Phi-3-mini-128k-instruct
microsoft/Phi-3-mini-4k-instruct
microsoft/Phi-3-small-8k-instruct |
|---|---|---|---|
| PhinD | Phind/Phind-CodeLlama-34B-v2 | PhindCL 34B v2 | Phind/Phind-CodeLlama-34B-v2 |
| CodeQwen | Qwen/CodeQwen1.5-7B

Qwen/CodeQwen1.5-7B-Chat | CodeQwen1.5 7B

CodeQwen1.5 7B Chat | Qwen/CodeQwen1.5-7B
Qwen/CodeQwen1.5-7B-Chat |
| Qwen | Qwen/Qwen2.5-Coder-32B-Instruct
Qwen/Qwen2.5-Coder-14B-Instruct
Qwen/QwQ-32B-Preview

Qwen/Qwen2-0.5B-Instruct

Qwen/Qwen2-1.5B-Instruct

Qwen/Qwen2-57B-A14B-Instruct
Qwen/Qwen2-7B
Qwen/Qwen2-7B-Instruct | QwenCoder2.5 32B Inst
QwenCoder2.5 14B Inst
QwQ 32B

Qwen2 0.5B Instruct
Qwen2 1.5B Instruct
Qwen2 57B-A14B Instruct
Qwen2 7B
Qwen2 7B Instruct | Qwen/Qwen2.5-Coder-32B-Instruct
Qwen/Qwen2.5-Coder-14B-Instruct
Qwen/Qwen2-0.5B-Instruct
Qwen/Qwen2-0.5B-Instruct
Qwen/Qwen2-1.5B-Instruct
Qwen/Qwen2-57B-A14B-Instruct
Qwen/Qwen2-7B
Qwen/Qwen2-7B-Instruct |
| Yi | 01-ai/Yi-1.5-34B-Chat
01-ai/Yi-1.5-6B-Chat
01-ai/Yi-1.5-9B-Chat | Yi1.5 34B Chat
Yi-1.5 6B Chat
Yi1.5 9B Chat | 01-ai/Yi-1.5-34B-Chat
01-ai/Yi-1.5-6B-Chat
01-ai/Yi-1.5-9B-Chat |
| DeepSeek | deepseek-ai/DeepSeek-R1
deepseek-ai/deepseek-V3 | DeepSeek R1
DeepSeek V3 | deepseek-ai/R1
deepseek-ai/DeepSeek-V3 |
| DeepSeek Coder | deepseek-ai/deepseek-coder-33b-base
deepseek-ai/deepseek-coder-33b-instruct
deepseek-ai/deepseek-coder-6.7b-base
deepseek-ai/deepseek-coder-6.7b-instruct
deepseek-ai/deepseek-coder-7b-base-v1.5
deepseek-ai/deepseek-coder-7b-instruct-v1.5
deepseek-ai/DeepSeek-Coder-V2-Lite-Base
deepseek-ai/DeepSeek-Coder-V2-Lite-Instruct | DSCoder 33B
DSCoder 33B Instruct
DSCoder 6.7B
DSCoder 6.7B Instruct
DSCoder 7B (v1.5)
DSCoder 7B Instruct (v1.5)
DSCoderV2 Lite
DSCoderV2 Lite Instruct | deepseek-ai/deepseek-coder-33b-base
deepseek-ai/deepseek-coder-33b-instruct
deepseek-ai/deepseek-coder-6.7b-base
deepseek-ai/deepseek-coder-6.7b-instruct
deepseek-ai/deepseek-coder-7b-base-v1.5
deepseek-ai/deepseek-coder-7b-instruct-v1.5
deepseek-ai/DeepSeek-Coder-V2-Lite-Base
deepseek-ai/DeepSeek-Coder-V2-Lite-Instruct |
| DeepSeek MoE | deepseek-ai/deepseek-moe-16b-base
deepseek-ai/deepseek-moe-16b-chat | DSMoE 16B
DSMoE 16B Chat | deepseek-ai/deepseek-moe-16b-base
deepseek-ai/deepseek-moe-16b-chat |

| InternLM | internlm/internlm2_5-20b-chat
internlm/internlm2_5-7b-chat | InternLM2.5 20B Chat
InternLM2.5 7B Chat | internlm/internlm2_5-20b-chat
internlm/internlm2_5-7b-chat |
|---|---|---|---|
| StarCoder2 | bigcode/starcoder2-15b-instruct-v0.1
bigcode/starcoder2-7b | StarCoder2 15B Instruct
StarCoder2 7B | bigcode/starcoder2-15b-instruct-v0.1
bigcode/starcoder2-7b |

## C.1 PROMPT LIBARARY

**Filtering prompts:** LLM-based filtering for ranking questions' completeness, coherence, and clarity.

---

**Quality filtering prompt:**

---

```
Please rate the following question based on three criteria, with a score from 1 to 5 for
↪ each criterion (where 1 is the lowest and 5 is the highest). No explanation needed:

  1.  Completeness:
  - Does the question stand alone and provide enough information independently?
  - Avoid including any images, links, or external references.
  2.  Coherence and Clarity:
  - Is the question phrased clearly, with proper grammar?
  - Is there any ambiguity or confusion in the wording?
  3.  Relevance:
  - Is the question directly related to software development or programming issues?
  - Does it involve technical challenges, concepts, or tools commonly used in software
↪ or programming?

Question:
"""{}"""
```

---

**Data creation prompts:** Prompt used for synthesis distractor for real-world task:

---

**Code Repair distractor creation prompts:**

---

After extracting statement from buggy version, we use LLMs to rewrite a new version of that statement. We command LLMs to assume the bug is located in the assigned line and their target is correct that line.
Here is the prompt:

```
        Given a buggy Python code snippet, you will be asked to debugging the code.
        ```
        def truncate_number(number: float) -> float:
            return number * (number % 1)
        ```
        Let assume the bug is located in this line:
        ```    return number * (number % 1)```
        Adjust this line in order to solve the bug.
        The re-written line must be syntactic correct, executable and wrapped in ``` ```
↪ brace.
        Don't give any details.
        ### Rewritten line:
        ```    return number % 1.0```

        Given a buggy Java code snippet, you will be asked to debugging the code.
        ```{code}```
        Let assume the bug is located in this line:
        ```{line}```
        Adjust this line in order to solve the bug.
        The re-written line must be syntactic correct, executable and wrapped in ``` ```
↪ brace.
        Don't give any details.
        ### Rewritten line:
```

We executing the problem with given test cases. Our target is to create reasonable false answer that would require deep interpretation. Follow by an LLMs based filter to pick from pool of negative answer the most likely able to solve the buggy problem. This result a set of confusing negative answer. Those reasonable false sample with executable (and if they can pass through few testcases) is golden negative answer.

---

*Fill in the blank distractor creation prompt:*

From correct solution from leetcode, we randomly mask a line/a block of code and generate false answer (for multiple choice) from LLMs:

```
Following this code:
{code}
I prepare some multiple choice questions answering
so i want to make small change on this line
but it still look true of this line : {line}
help me generate 3 version change in this code and each output should in ''' '''
↪ brace and code only.
Don't give any details
```

**Experimental prompts:**  Prompt used in CodeMMLU evaluation.

**Zero-shot prompts**

*General knowledge MCQ test set:*

```
The following are multiple-choice questions (with answers) about software
↪ development.

Question: {question}
{multiple_choices}

Answer:
```

*Code completion:*

```
The following are multiple-choice questions (with answers) about software
↪ development.

Question: {question}
{multiple_choices}

Answer:
```

*Fill in the blank:*

```
The following are multiple-choice questions (with answers) about a programming
↪ problem with an incomplete solution.

Problem statement: {question}

Incomplete Solution:
{codebase}

Question: The provided solution is missing a part, Which option below is the
↪ most likely to complete the solution and achieve the desired goal?

{multiple_choices}

Answer:
```

*Code Repair:*

```
        The following are multiple-choice questions (with answers) about debugging a
↪ programming problem.

        Question: The implementation below is producing incorrect results. Which
↪ solution below correctly identifies the bug and repairs it to achieve the desired
↪ goal?
        {question}

        {multiple_choices}

        Answer:
```

*Defect Detection:*

```
        The following are multiple-choice questions (with answers) about programming
↪ problems.

        Question: Given a code snippet below, which behavior most likely to occur when
↪ execute it?
        {question}

        {multiple_choices}

        Answer:
```

**Few-shot prompt**

*General knowledge MCQ test set:*

```
        The following are multiple choice questions (with answers) about software
↪ development.

        Question: If a sorted array of integers is guaranteed to not contain duplicate
↪ values, in order to search a for a specific value which of the following algorithms
↪ is the most efficient for this task?

        (A) Bubble Sort (B) Linear Search (C) Insertion Sort (D) Binary Search

        Answer: The answer is (D).

        Question: {question}
        {multiple_choices}

        Answer:
```

*Code completion:*

```
        The following are multiple-choice questions (with answers) about programming
↪ problems.

        Question: Which solution below is the most likely completion the following code
↪ snippet to achieve the desired goal?
        ```python
        from typing import List

        def two_sum(nums: List[int], target: int) -> List[int]:
            """
            Given an array of integers nums and an integer target, return indices of the
↪ two numbers such that they add up to target.
            You may assume that each input would have exactly one solution, and you may
↪ not use the same element twice.

            >>> two_sum([2,7,11,15], 9)
            [0,1]
            >>> two_sum([3,2,4], 6)
            [1,2]
            >>> two_sum([3,3], 6)
            [0,1]
            """
        ```

        (A) ```python
            n = len(nums)
            for i in range(n - 1):
                for j in range(i + 1, n):
                    if nums[i] + nums[j] == target:
                        return [i, j]
            return []
        ```
        (B) ```python
            for num in nums:
                if target - num in nums:
                    return [nums.index(num), nums.index(target - num)]
            return []
        ```
```

```python
(C) ```python
    for i in range(len(nums)):
        if nums[i] * 2 == target:
            return [i, i]
    return []
```
(D) ```python
    num_dict = {}
    for i, num in enumerate(nums):
        if target - num in num_dict:
            return [num_dict[target - num], i]
        num_dict[i] = num
    return []
```
```

Answer: The answer is A.

Question: Which solution below is the most likely completion the following code
↪ snippet to achieve the desired goal?
```python
{question}
```

{multiple_choices}

Answer:'''

---

*Fill in the blank:*

The following are multiple-choice questions (with answers) about a programming
↪ problem with incomplete solution.

Problem statement: You are given an array of intervals, where intervals[i] =
↪ [starti, endi] and each starti is unique.
The right interval for an interval i is an interval j such that startj >= endi
↪ and startj is minimized.
Note that i may equal j. Return an array of right interval indices for each
↪ interval i.
If no right interval exists for interval i, then put -1 at index i.

Incomplete Solution:
python```
def find_right_interval(intervals):
    n = len(intervals)
    res = [-1] * n
    for i in range(n):
        intervals[i].append(i)

    def binary_search(ele):
        left, right = 0, n-1
        ans = float('inf')
        while left <= right:
            mid = (left + right) // 2
            if intervals[mid][0] >= ele:
                ans = min(ans, mid)
                right = mid - 1
            else:
                left = mid + 1
        return ans

    intervals.sort()
    for i in intervals:
        -----------------

    return res
```

Question: The provided solution is missing a part, Which option below is the
↪ most likely to complete the solution and achieve the desired goal?

```
(A) ```python
    val = binary_search(i[1])
    if val != float('inf'):
        res[i[2]] = intervals[val][2]
```
(B) ```python
    if val != float('inf'):
        res[i[2]] = intervals[val][2]
    else:
        continue
```
(C) ```python
    val = binary_search(i[1])
    if val != float('inf'): res[i[2] + 1] = intervals[val][2]
```
(D) ```python
    if val != float('inf'):
        res[i[2]] = intervals[val][2]
    else:
        continue
```
Answer: The answer is (A).

Problem statement: {question}

Incomplete Solution:
{codebase}

Question: The provided solution is missing a part, Which option below is the
↪ most likely to complete the solution and achieve the desired goal?

{multiple_choices}

Answer:
```

*Code Repair:*

```
        The following are multiple-choice questions (with answers) about debugging a
↪ programming problem.

        Question: The implementation below is producing incorrect results.
        Which solution below correctly identifies the bug and repairs it to achieve the
↪ desired goal?

        1 def two_sum(nums, target):
        2     complement_map = {{}}
        3     for i, num in enumerate(nums):
        4         complement = target - num
        5         complement_map[num] = i
        6         if complement in complement_map:
        7             return [complement_map[complement], i]
        8     return None

        (A) Remove line 5.

        (B) Remove line 5. Add at line 7:
        ```            complement_map[num] = i```

        (C) Modify line 7:
        ```             return [i, complement_map[complement]]```

        (D) Remove line 5. Add at line 7:
        ```    if i == len(nums) - 1:
                    return None
                complement_map[num] = i```

        Answer: The answer is (B).

        Question: The implementation below is producing incorrect results.
        Which solution below correctly identifies the bug and repairs it to achieve the
↪ desired goal?
        {question}

        {choices}

        Answer:
```

*Defect Detection:*

```
        The following are multiple choice questions (with answers) about programming
↪ problem.

        Question: Given a code snippet below, which behavior most likely to occurr when
↪ execute it?
        ```python
        def chkPair(A, size, x):
            for i in range(0, size - 1):
                for j in range(i + 1, size):
                    if (A[i] + A[j] == x):
                        return 1
            return 0

        ```

        (A). The code contain no issue.
        (B). Memory Limit Exceeded
        (C). Compile error
        (D). Runtime Error
```

```
Answer: The answer is (A).

Question: Given a code snippet below, which behavior most likely to occurr when
↪ execute it?
{question}

{multiple_choices}

Answer:
```

---

**Chain-of-Thought zero-shot prompts**

---

*General knowledge MCQ test set:*

```
        The following are multiple choice questions (with answers) about software
↪  devopment.

        Question: {question}
        {multiple_choices}

        Answer: Let's think step by step.
```

---

*Code completion:*

```
        The following are multiple choice questions (with answers) about programming
↪  problems.

        Question: Which solution below is the most likely completion the following code
↪  snippet to achieve the desired goal?
        '''python
        {question}
        '''
        {multiple_choices}

        Answer: Let's think step by step.
```

---

*Fill in the blank:*

```
        The following are multiple-choice questions (with answers) about a programming
↪  problem with uncomplete solution.

        Problem statement: {question}

        Incomplete Solution:
        {codebase}

        Question: The provided solution is missing a part, Which option below is the
↪  most likely to
        complete the solution and achieve the desired goal?

        {multiple_choices}

        Answer: Let's think step by step.
```

---

*Code Repair:*

```
        The following are multiple-choice questions (with answers) about debugging a
↪  programming problem.

        Question: The implementation below is producing incorrect results.
        Which solution below correctly identifies the bug and repairs it to achieve the
↪  desired goal?
        {question}

        {multiple_choices}

        Answer: Let's think step by step.
```

---

*Defect Detection:*

```
        The following are multiple-choice questions (with answers) about debugging a
↪ programming problem.

        The algorithm implementation below is producing incorrect results;
        Which solution below correctly identifies the bug and repairs it to achieve the
↪ desired goal?
        {question}

        {multiple_choices}

        Answer: Let's think step by step.
```

**Chain-of-Thought few-shot prompts**

*General knowledge MCQ test set:*

```
        The following are multiple choice questions (with answers) about software
↪ devopment.

        Question: If a sorted array of integers is guaranteed to not contain duplicate
↪ values, in order to search a for a specific value which of the following algorithms
↪ is the most efficient for this task?

        (A) Bubble Sort (B) Linear Search (C) Insertion Sort (D) Binary Search

        Answer: Let's think step by step. Binary Search is a divide-and-conquer
↪ algorithm that works by repeatedly dividing the search interval in half and searching
↪ for the value in the appropriate half. Since the array is already sorted and does not
↪ contain any duplicate value, this algorithm is optimal to find the desired value. The
↪ answer is (D).

        Question: {question}
        {multiple_choices}

        Answer: Let's think step by step.
```

*Code completion:*

```
        The following are multiple choice questions (with answers) about programming
↪ problem.

        Question: Which solution below is the most likely completion the following code
↪ snippet to achieve the desired goal?
        ```python
        def is_vowel(char: str) -> bool:
            """
            Checks if the input character is a vowel.
            """
        ```

        (A) ```python
            return char.lower().is_vowel()
        ```
        (B) ```python
            vowels = set("aeiou")
            return char.lower() in vowels
        ```
        (C) ```python
            vowels = set("aeiou")
            return char.upper() in vowels
        ```
        (D) ```python
            vowels = "aeiou"
            return char.count(vowels) > 0
        ```

        Answer: Let's think step by step. The goal is to write a function is_vowel(char:
↪ str) -> bool that checks if the input character char is a vowel. The solution B
↪ correctly converts the input character to lowercase and checks if it is in the set of
↪ vowels.
        The answer is (B).

        Question: Which solution below is the most likely completion the following code
↪ snippet to achieve the desired goal?
        ```python
        {question}
        ```
        {multiple_choices}

        Answer: Let's think step by step.
```

*Fill in the blank:*

```
        The following are multiple-choice questions (with answers) about a programming
↪ problem with uncomplete solution.

        Problem statement: You are given an array of intervals, where intervals[i] =
↪ [starti, endi] and each starti is unique.
        The right interval for an interval i is an interval j such that startj >= endi
↪ and startj is minimized.
        Note that i may equal j. Return an array of right interval indices for each
↪ interval i.
        If no right interval exists for interval i, then put -1 at index i.

        Incomplete Solution:
        python''`
        def find_right_interval(intervals):
            n = len(intervals)
            res = [-1] * n
            for i in range(n):
                intervals[i].append(i)
            def binary_search(ele):
                left, right = 0, n-1
                ans = float('inf')
                while left <= right:
                    mid = (left + right) // 2
                    if intervals[mid][0] >= ele:
                        ans = min(ans, mid)
                        right = mid - 1
                    else:
                        left = mid + 1
                return ans

            intervals.sort()
            for i in intervals:
                -----------------

            return res
        ```

        Question: The provided solution is missing a part, Which option below is the
↪ most likely to
        complete the solution and achieve the desired goal?

        (A) ```python
            val = binary_search(i[1])
            if val != float('inf'):
                res[i[2]] = intervals[val][2]
        ```
        (B) ```python
            if val != float('inf'):
                res[i[2]] = intervals[val][2]
            else:
                continue
        ```
        (C) ```python
            val = binary_search(i[1])
            if val != float('inf'): res[i[2] + 1] = intervals[val][2]
        ```
        (D) ```python
            if val != float('inf'):
                res[i[2]] = intervals[val][2]
            else:
              continue
        ```
```

```
        Answer: Let's think step by step. The incomplete solution first sorts the
↪ intervals and then iterates over the sorted intervals. For each interval, it finds
↪ the right interval using a binary search.
        This option (A) finds the right interval index using the binary search and
↪ updates the result array accordingly.
        The option (B) is similar to (A), but it does not increment the index when
↪ finding the right interval index. This could lead to incorrect results.
        The option (C) increments the index when finding the right interval index.
↪ However, this is incorrect because the problem statement asks for the index of the
↪ right interval, not the offset from the original index.
        The option (D) uses the same index for both the original interval and the right
↪ interval, which could lead to incorrect results.
        The answer is (A).

        Problem statement: {question}

        Incomplete Solution:
        {codebase}

        Question: The provided solution is missing a part, Which option below is the
↪ most likely to
        complete the solution and achieve the desired goal?

        {multiple_choices}

        Answer: Let's think step by step.
```

***Code Repair:***

The following are multiple-choice questions (with answers) about debugging a
↪ programming problem.

Question: The implementation below is producing incorrect results.
Which solution below correctly identifies the bug and repairs it to achieve the
↪ desired goal?

```python
1 def two_sum(nums, target):
2     complement_map = {{}}
3     for i, num in enumerate(nums):
4         complement = target - num
5         complement_map[num] = i
6         if complement in complement_map:
7             return [complement_map[complement], i]
8     return None
```

(A) Remove line 5.

(B) Remove line 5. Add at line 7:
``` 
            complement_map[num] = i```

(C) Modify line 7:
```
            return [i, complement_map[complement]]```

(D) Remove line 5. Add at line 7:
```     if i == len(nums) - 1:
                return None
            complement_map[num] = i```

Answer: Let's think step by step. The bug in the code occurs because the current
↪ number is added to the complement_map before checking if its complement already
↪ exists, which can lead to incorrectly matching a number with itself. To fix this, the
↪ number should only be added to the map after checking for its complement. Solution
↪ (B) does exactly this by moving the line that adds the current number to the map
↪ after the complement check, ensuring the logic works as intended without
↪ self-matching errors.
The answer is (B).

Question: The implementation below is producing incorrect results.
Which solution below correctly identifies the bug and repairs it to achieve the
↪ desired goal?
{question}

{choices}

Answer: Let's think step by step.

*Defect Detection:*

```
    The following are multiple choice questions (with answers) about programming
↪ problem.

    Question: Given a code snippet below, which behavior most likely to occurr when
↪ execute it?
    ```python
    def chkPair(A, size, x):
        for i in range(0, size - 1):
            for j in range(i + 1, size):
                if (A[i] + A[j] == x):
                    return 1
        return 0

    ```
    (A). The code contain no issue.
    (B). Memory Limit Exceeded
    (C). Compile error
    (D). Runtime Error

    Answer: Let's think step by step. The code appears to have no issues with
↪ typical valid inputs and will function as expected. It correctly checks for pairs of
↪ elements whose sum is x.
    The answer is (A).

    Question: Given a code snippet below, which behavior most likely to occurr when
↪ execute it?
    {question}
    {multiple_choices}

    Answer: Let's think step by step.
```