# OpenReview forum: "CodeMMLU: A Multi-Task Benchmark for Assessing Code Understanding & Reasoning Capabilities of CodeLLMs"
_ICLR.cc/2025/Conference — ICLR 2025 Poster_

### Official Review · Reviewer_wsRv · 2024-10-22

**Soundness:** 3
**Presentation:** 2
**Contribution:** 3
**Rating:** 8
**Confidence:** 4

**Summary:**

The authors present CodeMMLU, a multiple-choice question-answer benchmark for code, consistenting of thousands of code/software-related questions, spanning syntactic knowledge (e.g., API and Frameworks, PL syntax), semantic knowledge (e.g., OOP, compiler design), and real-world tasks (e.g., code completion, code repair). This is inspired by the MMLU evaluation set used in NLP as well as programmer comprehension behavior models (Shneiderman & Mayber, 1979). The knowledge-based tasks are derived from W3Schools, Geeks4Geeks, and Sanfoundry, with LLM-based filtering. The real-world tasks are derived by re-purposing existing evaluation sets, with additional steps for synthesizing distractors with LLMs and execution-based filtering. The authors benchmark a large number of closed-source and open-source LLMs with CodeMMLU.

**Strengths:**

- This is a very large and diverse evaluation set, spanning many different topics, concepts, tasks, and programming languages. This could potentially be useful for the research community, providing a way forward for hill-climbing to improve code understanding in LLMs.
- The authors evaluate several open-source and closed-source code LLMs on this new benchmark, reporting numbers separately for syntactic, semantic, and real-world tasks. This is useful for understanding the weaknesses of current models, also paving a path forward for improvement.
- I particularly like the way the authors have generated hard distractors for the real-world tasks. This makes the task more difficult.
- There are some key insights from this work that are quite interesting. For instance, the authors show that CoT prompting is often not effective for CodeMMLU tasks, whereas few-shot prompting seems to consistently perform well. Additionally, the authors compare performance of HumanEval as a generative task versus as a MCQA task, through which they show that MCQA performance can be much lower sometimes. This suggests that generative tasks do not adequately evaluate a model's code reasoning capabilities.

**Weaknesses:**

- **Not clear whether this is a reliable evaluation set.** The correlation with human judgement has not been measured. The authors motivate this work by highlighting the issues of potential data leakage with existing benchmarks (L036). However, it seems that CodeMMLU is susceptible to the same issue. Data sources like W3Schools, Geeks4Geeks, and Sanfoundry are likely already in the pretraining sets of existing models. Additionally, the real-world tasks are based on existing benchmarks, which have leakage issues, as the authors claimed. Next, Figure 7 and Table 4 suggest that the performance is very sensitive to the position of the correct option, which suggests that there are factors beyond code comprehension at play in MCQA. Therefore, it is not clear whether we can rely on this for evaluation code comprehension.


- **Many missing details and also some details which are inconsistent.** First, it is not clear how the authors generated MCQA questions and hard alternative options for the knowledge-based tasks. Next, 10K is approximation given in the abstract for the number of examples in CodeMMLU. However, the sum across subjects in Table 2 is 20, 281. Does that mean there are some duplicates? Furthermore, the number of models that have been benchmarked is not clear. In Section 4.1, the authors say 35 open-source models (L312) and 3 closed-source models (L319). However, the number of rows in Table 3 do not align with this. In L375, the authors say they have evaluated 43 models across 10 model families. In L976, the authors say they have experimented on 48 LLM models from over 12 families. Additionally, some of the results are difficult to interpret. For example, there is no y-axis for Figure 5 and also the prompting style is not actually labeled in Figure 9.


**Suggestions**:
- Is 3.2 mis-labeled? Should it correspond to "Real-world problem set"
- Place Table 3 before Figure 4.
- Currently, Figure 5 is referred to before Figure 4. Maybe switch the numbering?
- L345: Detaill $\rightarrow$ Detail
- Table 3 is confusing. CodeMMLU is the aggregate score across Syntacic, Semantic, and Real-world tasks? Make this clear by saying "Combined" instead. CodeMMLU includes all.
- L426: There is no Table 8. Seems that the authors intended Figure 7.

**Questions:**

Please address the concerns raised in the Weaknesses section.

---

> ### Author Response · Authors · 2024-11-23
>
> **Q1: Evaluation reliability**
>
> > Not clear whether this is a reliable evaluation set. The correlation with human judgement has not been measured.
> >
>
> We want to clarify that our dataset is both reliable and trustworthy, as it is constructed using rigorous filtering and validation procedures. Firstly, we source data from widely recognized sources for programming knowledge (e.g., W3School, CommonCrawl) and research works, which have already undergone measurement and validation. Secondly, for the real-world tasks, we validate both the correct answers and the distractors through execution, ensuring their correctness. Finally, we manually verified a small subset of the data to confirm the reliability of the construction process. While directly measuring the correlation with human judgment across 20,000 examples would provide additional insights, it is practically infeasible and prohibitively expensive for such a large-scale evaluation.
>
> **Q2: Data leakage issues**
>
> > The authors motivate this work by highlighting the issues of potential data leakage with existing benchmarks (L036). However, it seems that CodeMMLU is susceptible to the same issue. Data sources like W3Schools, Geeks4Geeks, and Sanfoundry are likely already in the pretraining sets of existing models.
> >
> >
> > Additionally, the real-world tasks are based on existing benchmarks, which have leakage issues, as the authors claimed.
> >
>
> For clarification, we mitigated this issue by implementing rigorous filtering processes to ensure high-quality data. A key aspect of CodeMMLU’s design is the reformulation of raw data into the multiple-choice question (MCQ) format (as detailed in Section 3.2), which involves generating synthetic distractors as incorrect options. This transformation can reduce the likelihood that the questions in CodeMMLU have been encountered by LLMs during training, as LLMs are predominantly trained on raw code, bug reports, and similar data sources.
>
> **Table 1.** `ppl` of benchmarks *(higher is better)*
>
> | **Models** | **CodeScope** | **CodeApex** | **CodeMMLU** |
> | --- | --- | --- | --- |
> | mistralai/Mistral-7B-v0.3 | 9.315170 | 16.08231 | **16.31779** |
> | deepseek-ai/deepseek-coder-7b-base-v1.5 | 5.25711 | 9.39178 | **57.36** |
> | deepseek-ai/DeepSeek-V2-Lite | 6.889910 | 11.98695 | **1419.4829** |
> | meta-llama/Llama-3.1-8B | 10.05143 | 123.2007 | **197.30578** |
>
> **Table 2.** `5-gram` of benchmarks *(lower is better)*
>
> | **Models** | **CodeScope** | **CodeApex** | **CodeMMLU** |
> | --- | --- | --- | --- |
> | mistralai/Mistral-7B-v0.3 | 0.250963 | 0.1702479 | **0.13652** |
> | deepseek-ai/deepseek-coder-7b-base-v1.5 | 0.281777 | 0.168044 | **0.14157** |
> | deepseek-ai/DeepSeek-V2-Lite | 0.249245 | 0.15867768 | **0.068664** |
> | meta-llama/Llama-3.1-8B | 0.221852 | 0.130854 | **0.065229** |
>
> To further measure the degree of data leakage in benchmarks, we adopted the methodology from BenBench [1], utilizing perplexity and n-gram metrics. As shown in Table 1 and Table 2 (and in Appendix A.3 of the revision), CodeMMLU demonstrates lower levels of data leakage—evidenced by higher perplexity and lower n-gram overlap—compared to existing benchmarks like CodeScope and CodeApex. These results highlight the effectiveness of CodeMMLU’s pre-processing pipeline in mitigating data leakage.
>
> **Q3: MCQ biased and untrustworthy.**
>
> > Next, Figure 7 and Table 4 suggest that the performance is very sensitive to the position of the correct option, which suggests that there are factors beyond code comprehension at play in MCQA. Therefore, it is not clear whether we can rely on this for evaluation code comprehension.
> >
>
> Thank you for raising this concern. We believe the factors mentioned stem from the inherent bias problems of LLMs, which have also been extensively discussed in the context of MCQ evaluation in the NLP domain [1]. We want to clarify that this issue does not undermine the validity of evaluating code comprehension using MCQs but rather highlights a limitation in the current capabilities of LLMs. Addressing such biases is a challenge that ultimately lies within the responsibility of model developers. We refer reviewers to appendix B.1 in revision and the discussion of MCQs bias and their evidence report in Q3 of reviewer xQZ6.

---

> > ### Author Response · Authors · 2024-11-23
> >
> > **Q4: Confusing information**
> >
> > > It is not clear how the authors generated MCQA questions and hard alternative options for the knowledge-based tasks.
> > >
> >
> > > Furthermore, the number of models that have been benchmarked is not clear. In Section 4.1, the authors say 35 open-source models (L312) and 3 closed-source models (L319). However, the number of rows in Table 3 do not align with this.
> > >
> >
> > > In L375, the authors say they have evaluated 43 models across 10 model families. In L976, the authors say they have experimented on 48 LLM models from over 12 families.
> > >
> >
> > > Next, 10K is an approximation given in the abstract for the number of examples in CodeMMLU. However, the sum across subjects in Table 2 is 20,281.
> > >
> >
> > We appreciate your concern about the CodeMMLU knowledge test sets. The knowledge-based distractors are collected in the process of constructing the test set along with their questions; we did not synthesize their false answers. We acknowledge the typographical mistake in the main paper; in our new revision, we report all model descriptions that have been used for the experiment and study in appendix C. We also revised the data construction of knowledge-based (section 3.1, main paper) and addressed all model setup inaccurate information.
> >
> > **Q5: Writing improvement**
> >
> > > Additionally, some of the results are difficult to interpret. For example, there is no y-axis for Figure 5 and also the prompting style is not actually labeled in Figure 9.
> > >
> >
> > Thank you for pointing out this issue. We acknowledge the difficulty in interpreting some of our figure. We addressed your concern in our rebuttal revision.
> >
> > ---
> > [1] Zheng, C., Zhou, H., Meng, F., Zhou, J., & Huang, M. (2023, September). Large language models are not robust multiple choice selectors. In *The Twelfth International Conference on Learning Representations*.

---

> > > ### Comment · Reviewer_wsRv · 2024-11-25
> > > **Updating Score**
> > >
> > > Thank you for your response and for your changes to the manuscript. I have increased my score.

---

> ### Author Response · Authors · 2024-11-26
>
> Dear Reviewer wsRv26,
>
> We are delighted that our responses addressed your concerns. We thank the Reviewer for carefully assessing our work and adjusting the score accordingly after the discussion.
>
> Best,
>
> Authors

---

### Official Review · Reviewer_6QYn · 2024-10-22

**Soundness:** 2
**Presentation:** 3
**Contribution:** 3
**Rating:** 6
**Confidence:** 4

**Summary:**

This paper proposes CodeMMLU, a benchmark for evaluating LLMs in code understanding using multiple-choice questions. CodeMMLU consists of a group of different tasks across knowledge-based tests and real-world programming questions. The authors evaluated 35 LLMs on the CodeMMLU benchmark, and the results suggests that the performances of LLMs on CodeMMLU is not always consistent with the performances on code generation benchmarks.

**Strengths:**

+: The authors proposed a new benchmark in code understanding, which is long ignored in LLM for code evaluation.

+: CodeMMLU consists of a wide variety of code comprehension tasks, from syntax/semantic understanding to code repair and defect prediction.

+: The authors conducted extensive experiments on CodeMMLU with various LLMs.

**Weaknesses:**

-: For most tasks, the authors use LLMs to generate distractors. The quality of these generated distractors should be discussed.

-: The code completion and fill-in-the-blank tasks are more related to code generation instead of code understanding. Especially, the code completion task is based on the existing HumanEval dataset.

**Questions:**

- The decline in accuracy with COT prompts is interesting. Perhaps it's better to analyze the LLMs' answers with COT in detail.

- In section 3.2, why is predicting execution output under the same category as defect prediction?

---

> ### Author Response · Authors · 2024-11-23
>
> We extend our sincere gratitude for your valuable insights and thoughtful evaluation of our work.
>
> **Q1: Distractor quality concern**
>
> > -: For most tasks, the authors use LLMs to generate distractors. The quality of these generated distractors should be discussed.
> >
>
> Thank you for your valuable suggestion. We have provided detailed explanations of how distractors are constructed for each subject in Sections 3.2, 3.3 and Appendix A1. Specifically, these distractors are generated using LLMs (Mistral 8x7B Instruct, GPT-3.5)  and designed to appear plausible while being intentionally incorrect. To ensure their validity as incorrect answers, we verify them through an execution-based process, confirming that they are executable but do not lead to the correct solution (we kept those that met the threshold of lower than 50% test cases passed). This approach helps maintain the quality and challenge of the MCQs in our benchmark.
>
> **Q2: Concern about Code understanding task**
>
> > -: The code completion and fill-in-the-blank tasks are more related to code generation instead of code understanding. Especially, the code completion task is based on the existing HumanEval dataset.
> >
>
> We agree that code completion and fill-in-the-blank tasks are more closely related to code generation. However, in our benchmark, we reformulate these tasks into an MCQ format, requiring models to comprehend the questions and options to select the correct answer. For the code completion task, we selected HumanEval due to its simplicity. Nevertheless, our MCQ formulation introduces new challenges that go beyond its generative format, as evidenced by the performance drop in Table 4 (main paper) and the misalignment of solved problems between the two formats shown in Figure 7. These results help us to highlight reasoning and comprehension weaknesses in LLMs, which are not effectively captured by generation-based benchmarks, even relatively simple ones like HumanEval.
>
> **Q3: Analysis CoT and other prompt settings**
>
> > The decline in accuracy with COT prompts is interesting. Perhaps it's better to analyze the LLMs' answers with COT in detail.
> >
>
> **Answer:**
>
> Thank you for your insightful suggestion regarding the detailed analysis of CoT (Chain-of-Thought) results. To address this concern, we have added a specific example of CoT usage and included a performance comparison against zero-shot prompting in Appendix B2. Additionally, we have updated the complete experimental results for all prompt settings studied in the paper in Appendix B3.
>
> For a focused study, we selected the Object-Oriented Programming subset—a smaller subject in CodeMMLU’s knowledge test set consisting of 64 questions—to evaluate the effectiveness of the CoT technique. Our findings align with the conclusions of Sprague et al. [1], which suggest that while CoT introduces additional reasoning steps, these steps can either assist in overcoming challenges or inadvertently make the questions more complex due to added distractions. In our experiments, CoT did not exhibit a consistent pattern of overcoming new challenges, regardless of whether short or long prompts were employed, highlight reasoning might not always yield a clear advantage against knowledge-seeking tasks, such as MMLU and CodeMMLU.
>
> **Q4: Name confusion**
>
> > In section 3.2, why is predicting execution output under the same category as defect prediction?
> >
>
> We acknowledge that the name "defect detection" may have been misleading and have updated it to "Execution Prediction" in Section 3.2 of our revision.
>
>
> [1] Sprague, Z., Yin, F., Rodriguez, J. D., Jiang, D., Wadhwa, M., Singhal, P., … & Durrett, G. (2024). To cot or not to cot? chain-of-thought helps mainly on math and symbolic reasoning. *arXiv preprint arXiv:2409.12183*.

---

> ### Author Response · Authors · 2024-11-26
>
> Dear Reviewer **6QYn,**
>
> Thank you for your time and valuable feedback. We hope our previous response has adequately addressed your concerns regarding the CodeMMLU sources and filtering quality. We eagerly await your feedback and are happy to clarify any additional questions.
>
> Best,
>
> Authors

---

> > ### Author Response · Authors · 2024-11-29
> >
> > Dear Reviewer **6QYn**,
> >
> > We hope our previous response has adequately resolved your questions or concerns.
> > As the deadline for the ICLR rebuttal period is approaching, we look forward to hearing your feedback on our response, and would be pleased to clarify any additional questions.
> >
> > Best,
> >
> > Authors

---

### Official Review · Reviewer_aVYU · 2024-11-03

**Soundness:** 2
**Presentation:** 3
**Contribution:** 2
**Rating:** 5
**Confidence:** 4

**Summary:**

his paper presents CodeMMLU, a collection of almost 20,000 multiple choice questions and answers on coding, collected from the web.

The paper covers three areas of related work: benchmarks for program comprehension, models for program comprehension, and multi-task question/answering benchmarks.

The paper collects programming-relaed multiple-choice questions from web sites like GeeksForGeeks, W3Schools, and Sanfoundry. The questions cover syntactic (programming languages and API) and semantic (algorithms, design, OO) knowledge, as well tasks (code completion, fill-in-the-blank, defect detection, and code repair).

These 20,000 questions were evaluated on 35 "open source" code LLMs.
Key reported insights include:

- Performance on knowledge questions and task questions is correlated
- LLM's preference for certain answers (e.g., avoid 'A') in multiple-choice questions is also present in code models
- Chain-of-Thought is not helpful for these questions
- Turning HumanEval into multiple choice questions changes model performance

**Strengths:**

The dataset is impressive, with almost 20,000 questions, and represents a substantial amount of (manual) work

The grouping of questions is meaningful

The paper is easy to read and follow

The dataset puts code language model performance in a different perspective.

**Weaknesses:**

While I liked the paper, my main concerns are:

- While filters and manual processing are applied, these are described only very briefly. As a result, the quality of the questions is unclear. It appears the paper favors quantity over quality. Fewer questions but of guaranteed quality would have been better.

- The questions for the 'real world tasks', e.g., for defect detection or line completion, are very artificial

- The treatment of multiple-choice bias (models prefering to avoid option A) in the paper is unsatisfactory

- It is unclear to what extent the LLMs were exposed to the underlying resources for this new data set (leakage). This risk is not discussed nor mitigated.


The writing and presentation are generally good, yet is sloppy at places (the abstract speaks about "over 10,000" questions -- there are 19,900, which is more like 20,000, 3.2 speaks about "five distinct" tasks, but there are four, there is no table 8 (only a figure 8), ...). It is confusing that the text summarizing table 2 gives very different numbers from what is in the table ('over 3000' when in the table it appears to be closer to 5000, and 6000 when it is in fact 7000). I'm not sure why section A.3 is entitled "visualization" (nothing is visualized -- examples are given).

The filtering process is described, but the exact numbers involved (before/after filtering) are not provided. The filtering involves various manual steps -- applied to how many cases? Deep learners are used here, but no details are provided.

Referring to the tasks as "real world performance" is misleading. The tasks are still highly artificial. Concerning the tasks themselves:

- I was surprised to see 'code completion' tasks based on HumanEval -- HumanEval suffers from many problems. There is a vast amount of literature on LLM assisted code-completion using data that is better than HumanEval.
- The defect detection task appears to be about predicting the correct execution result -- which is a different task from defect detection. Again, there are lots of defect benchmarks around, with real bugs collected from actual systems (e.g., defects4j). It is not so clear what these mutliple choice questions add to that, especially with the weak distractros (like compile error, "internal error" (??))

The reporting of the results about preference for option A (figure 7, table 4) is very minimalistic. The bias is stated, but not really studied / explained, nor are mitigation measures such as proposed by Zheng et al applied. The paper writes that 'we experimented' with multiple answer order, but what exactly was done is unclear.
I must say these findings also undermine the whole endeavour. If the multiple choice format itself is a problem, what is the point of having a large multiple choice data set?

It is unclear how the dataset will be distributed. I would believe some of the data is copyright protected (e.g., W3Schools). This would mean you cannot redistribute their multiple choice questions.

**Questions:**

Do all questions have four alternatives (one correct and 3 distractors)? At any rate, a random guesser would get 25% right, which makes the results in Table 3 less impressive.

Were the questions or the sources they were derived from included in the training data of the language models benchmarked? How are such risks mitigated?

Can you explain the "CodeMMLU" column in Table 3? I thought it was the overall performance, but it's not explained. How can the Meta-Llama-3.170B-Instruct overall be 60, while the other three columns are all above 64? (Also please align around decimal point instead of centering numeric columns).

Can you explain how you will distribute the dataset and under what licence the original material was made available, and which license you intend to use?

**Details Of Ethics Concerns:**

It is unclear how the dataset will be distributed. I would believe some of the data is copyright protected (e.g., W3Schools). This would mean you cannot redistribute their multiple choice questions. Please explain.

---

> ### Author Response · Authors · 2024-11-23
>
> We genuinely value your thoughtful feedback and constructive questions.
>
> **Q1: Benchmark license and distributing concern**
>
> It is unclear how the dataset will be distributed. I would believe some of the data is copyright protected (e.g., W3Schools). This would mean you cannot redistribute their multiple choice questions.
>
> Can you explain how you will distribute the dataset and under what licence the original material was made available, and which license you intend to use?
>
> We thank the Reviewer for raising this critical concern. In short, we will release CodeMMLU under the MIT license.
>
> We construct CodeMMLU by curating data from the web, most of which are from the Common Crawl, and thus can be used for academic purposes. For data crawled from websites such as W3schools and Geeks4geeks, we fully complied with their copyrights or sought their permission to use such data for this project. Thus, the MIT license satisfies all the sources’s copyrights.
>
> Lastly, we wish to clarify a typo we made in the initial submission, where the category “programming language syntax” of the knowledge task was collected from CC but was annotated as “sanfoundry”.  We double-checked the license of all data and fixed this typo. For a detailed breakdown of the licensing from each source, we refer the Reviewers to appendix A.3, revision version.
>
> **Q2: Filtering and processing quality concern**
> > While filters and manual processing are applied, these are described only very briefly. As a result, the quality of the questions is unclear.
> >
>
> We acknowledge that the description of our filtering and manual processing methods could be more detailed. We have addressed this by expanding Section 3.3 and A.1 in the revision to provide a clearer explanation of our quality assurance process. (we refer reviewers to discussion Q6 of reviewer xQZ6)
>
> Specifically, we employ a rule-based filtering approach to remove data with incomplete information, non-textual content, or irrelevant and redundant samples. Additionally, we utilize LLMs to assess and filter questions based on completeness, coherence, clarity, and relevance. To ensure the dataset remains challenging, we also use LLMs to evaluate and exclude overly simplistic questions based on difficulty. This combined approach ensures that while our dataset is extensive, it maintains a high standard of quality and relevance. Finally, we manually review of a small subset of the data (100 instances per subject) to guarantee the effectiveness and quality of the automated filtering processes.
>
> **Q3: Real-world concerning on real-world subset**
> > The questions for the 'real world tasks', e.g., for defect detection or line completion, are very artificial. [..] Referring to the tasks as "real world performance" is misleading.
> >
>
> > [..] 'code completion' tasks based on HumanEval [..] (which) suffers from many problems. There is a vast amount of literature on LLM assisted code-completion using data that is better than HumanEval.
> >
> >
> > The defect detection task appears to be about predicting the correct execution result -- which is a different task from defect detection. Again, there are lots of defect benchmarks around, with real bugs collected from actual systems (e.g., defects4j). It is not so clear what these mutliple choice questions add to that, especially with the weak distractros (like compile error, "internal error" (??))
> >
>
> We want to clarify that all samples in CodeMMLU are presented in a multiple-choice question (MCQ) format, which, while not explicitly present in practical software engineering tasks, provides a structured way to evaluate comprehension and decision-making. The task names, such as "real-world tasks," are suggestive and refer to the transformation of practical code scenarios (such as generation or debugging) into the MCQ format for evaluation purposes. Although these MCQs are abstractions, implicit decision-making scenarios akin to MCQs are part of a developer's daily work, such as choosing between implementation strategies or configuring environments.
>
> Regarding the use of HumanEval, we would appreciate it if the reviewer could elaborate on the specific concerns with HumanEval and suggest datasets they consider superior. We agree that HumanEval is relatively simple; our MCQs present new challenges that extend beyond its generative format. These challenges highlight reasoning and understanding weaknesses in LLMs, which may not be captured effectively by generation-based benchmarks.
>
> We acknowledge that the names "defect detection" and “real-world task” may have been misleading and have updated them to "Execution Prediction" and “Fundamental coding skill test" respectively, in Section 3.2 of our revised manuscript.

---

> ### Author Response · Authors · 2024-11-23
>
> **Q4: MCQ biases treatment**
>
> > The treatment of multiple-choice bias (models prefering to avoid option A) in the paper is unsatisfactory.
> > The bias is stated, but not really studied / explained, nor are mitigation measures such as proposed by Zheng et al applied. [..] If the multiple choice format itself is a problem, what is the point of having a large multiple choice data set?
> >
>
> We appreciate the reviewer’s concern regarding the multiple-choice biases in our work. However, our primary objective is not to solve the bias issue but to bring attention to and investigate the challenges that MCQs in the coding domain present to LLMs. Hence, the responsibility for addressing bias should be done by the model builders.
>
> While biases in MCQs have been observed in LLMs for natural language processing (NLP) [1], their manifestation in programming languages remains underexplored. Our results, as shown in Figure 7 and Table 4, highlight significant biases in LLMs when answering MCQs in the programming language domain, with the bias being even more pronounced than in NLP tasks. We disagree with the statement that the multiple-choice format itself is inherently problematic; rather, the issue lies in the capability of current LLMs. The use of a large-scale dataset like CodeMMLU allows us to demonstrate that these observations are not skewed by a small corpus, further underscoring the limitations of LLMs. Addressing these challenges will require advancements in model robustness and comprehension, which we hope our benchmark will help to inspire.
>
> We refer the reviewers to appendix B1 and response Q3 of reviewer xQZ6 for further discussion on MCQ bias.
>
> **Q5: Ambiguous filtering processing**
> > The filtering process is described, but the exact numbers involved (before/after filtering) are not provided. The filtering involves various manual steps -- applied to how many cases? Deep learners are used here, but no details are provided.
> >
>
> Thank you for your suggestion. We have expanded the description of the filtering process in Section 3.2 and Appendix A.2 to include detailed explanations of both the rule-based and deep learning approaches. Besides, we manually review a small subset of the data (100 instances per subject) to guarantee the effectiveness and quality of the automated filtering processes. For further clarification, please refer to our response to Q1. Additionally, we have updated the manuscript to include the result of filtering process (Figure 8 in Section A.1).
>
> **Q5: Improve writing and misleading information**
>
> > The writing and presentation are generally good, yet is sloppy at places (the abstract speaks about "over 10,000" questions -- there are 19,900, which is more like 20,000, 3.2 speaks about "five distinct" tasks, but there are four, there is no table 8 (only a figure 8), ...).
> It is confusing that the text summarizing table 2 gives very different numbers from what is in the table ('over 3000' when in the table it appears to be closer to 5000, and 6000 when it is in fact 7000). I'm not sure why section A.3 is entitled "visualization" (nothing is visualized -- examples are given).
> >
>
> We sincerely appreciate your detailed observations and have addressed and corrected these errors in the revised version. We have updated the abstract to reflect the correct number of samples and addressed writing issues in Section 3.2. Additionally, we have relocated Section A.3 to B.1, included examples of CodeMMLU.

---

> ### Author Response · Authors · 2024-11-23
>
> **Q7: Data leakage not discussed**
>
> > It is unclear to what extent the LLMs were exposed to the underlying resources for this new data set (leakage). This risk is not discussed nor mitigated.
> >
>
> We agree that data leakage is a critical issue to address when constructing a benchmark and we need to provide more discussion. For clarification, we mitigated this issue by implementing rigorous filtering processes to ensure high-quality data. A key aspect of CodeMMLU’s design is the reformulation of raw data into the multiple-choice question (MCQ) format (as detailed in Section 3.2), which involves generating synthetic distractors as incorrect options. This transformation can reduce the likelihood that the questions in CodeMMLU have been encountered by LLMs during training, as LLMs are predominantly trained on raw code, bug reports, and similar data sources.
>
> **Table 1.** `ppl` of benchmarks *(higher is better)*
>
> | **Models** | **CodeScope** | **CodeApex** | **CodeMMLU** |
> | --- | --- | --- | --- |
> | mistralai/Mistral-7B-v0.3 | 9.315170 | 16.08231 | **16.31779** |
> | deepseek-ai/deepseek-coder-7b-base-v1.5 | 5.25711 | 9.39178 | **57.36** |
> | deepseek-ai/DeepSeek-V2-Lite | 6.889910 | 11.98695 | **1419.4829** |
> | meta-llama/Llama-3.1-8B | 10.05143 | 123.2007 | **197.30578** |
>
> **Table 2.** `5-gram` of benchmarks *(lower is better)*
>
> | **Models** | **CodeScope** | **CodeApex** | **CodeMMLU** |
> | --- | --- | --- | --- |
> | mistralai/Mistral-7B-v0.3 | 0.250963 | 0.1702479 | **0.13652** |
> | deepseek-ai/deepseek-coder-7b-base-v1.5 | 0.281777 | 0.168044 | **0.14157** |
> | deepseek-ai/DeepSeek-V2-Lite | 0.249245 | 0.15867768 | **0.068664** |
> | meta-llama/Llama-3.1-8B | 0.221852 | 0.130854 | **0.065229** |
>
> To further measure the degree of data leakage in benchmarks, we adopted the methodology from BenBench [1], utilizing perplexity and n-gram metrics. As shown in the table 1 and table 2 (and in Appendix A.2 of the revision), CodeMMLU demonstrates lower levels of data leakage—evidenced by higher perplexity and lower n-gram overlap—compared to existing benchmarks like CodeScope and CodeApex. These results highlight the effectiveness of CodeMMLU’s pre-processing pipeline in mitigating data leakage.
>
> ---
> [1] Zheng, C., Zhou, H., Meng, F., Zhou, J., & Huang, M. (2023, September). Large language models are not robust multiple choice selectors. In *The Twelfth International Conference on Learning Representations*.

---

> ### Author Response · Authors · 2024-11-26
>
> Dear Reviewer **aVYU,**
>
> Thank you for your thoughtful comments and the time you have dedicated to our work. We hope our previous response has sufficiently addressed your concerns regarding the CodeMMLU license, data and filtering quality, and solutions for MCQ bias. We look forward to hearing your thoughts on our response and are happy to provide further clarification if needed.
>
> Best,
>
> Authors

---

> > ### Author Response · Authors · 2024-11-29
> >
> > Dear Reviewer **aVYU**,
> >
> > We hope our previous response has adequately resolved your questions or concerns.
> > As the deadline for the ICLR rebuttal period is approaching, we look forward to hearing your feedback on our response, and would be pleased to clarify any additional questions.
> >
> > Best,
> >
> > Authors

---

### Official Review · Reviewer_xQZ6 · 2024-11-08

**Soundness:** 2
**Presentation:** 1
**Contribution:** 2
**Rating:** 3
**Confidence:** 5

**Summary:**

The authors propose CodeMMLU, a new benchmark designed specifically to evaluate LLMs' code understanding capabilities. This benchmark includes over 10,000 questions covering various domains and programming languages. The question types include knowledge-based evaluations (such as programming language syntax, API usage, and software development principles) and practical programming tasks (such as code completion, fixing, and defect detection). The authors test various LLMs using this benchmark and provide insights into model performance, prompting strategies, and their correlation with practical programming skills.

**Strengths:**

1. CodeMMLU abandons traditional code generation evaluation methods and adopts a multiple-choice question format, shifting the focus from code generation to code understanding to evaluate LLMs' code understanding abilities.

2. The CodeMMLU benchmark includes over 10,000 diverse questions with broad coverage and high-quality sources (such as GeeksforGeeks, LeetCode, etc.). The authors have put in substantial work overall.

3. The authors conduct extensive experiments on CodeMMLU, providing experimental insights into multiple aspects such as selection bias in multiple-choice questions and correlations between LLMs' software knowledge and real-world applications. The authors also provide numerous tables, figures, and other visualizations to help readers understand the paper.

**Weaknesses:**

1. In the Related Work section, the authors simply list some code evaluation benchmarks without clearly articulating the specific differences between CodeMMLU and each existing benchmark, nor do they explain what issues CodeMMLU addresses that current benchmarks do not. In contrast, there are many comprehensive and thorough benchmarks for code understanding and code generation, such as CodeXGlue, XLCoST, xCodeEval, CodeScope, LiveCodeBench, and BigCodeBench. Notably, CodeScope (ACL 2024) has constructed a code understanding benchmark that includes four tasks and multiple-choice questions. Additionally, LiveCodeBench addresses the issue of data leakage through dynamic data set updates. CodeMMLU lacks detailed and thorough comparative analysis of key related works (a good example of related work analysis can be found in ClassEval). Moreover, the paper does not offer any solutions to the data leakage problem mentioned in line 36. Overall, although the paper involves considerable effort in data labeling and other aspects, it lacks novelty, with many conclusions already available in previous literature. The innovation and actual contributions of the paper remain unclear.

2. In paper, the authors discuss how LLMs are sensitive to the order of options in multiple-choice questions, which can lead to fluctuations in performance and thereby affect the accuracy of the results. Although the issue is acknowledged, unfortunately, no strategies for eliminating this bias are provided. This raises concerns about the reliability and robustness of benchmark tests. Moreover, the paper "Large Language Models Are Not Robust Multiple Choice Selectors," published at ICLR 2024, has demonstrated that using multiple-choice questions to evaluate LLMs is not stable and introduces significant biases. Given this, I am curious about how the authors address this issue.

3. The overall writing of this paper needs improvement, particularly in the areas of data handling and presentation where essential detailed explanations are lacking. Specifically, in Figure 3, titled "Overview of the CodeMMLU Data Creation Pipeline", the authors fail to clearly explain the process of constructing multiple-choice questions from various data forms after filtering. Moreover, the inclusion of the LLMs evaluation in Figure 3 is not well-explained. In line 319 of the text, although four models "GPT-3.5, GPT-4, Claude-3-opus, and Claude-3.5-sonnet" are tested, the manuscript inaccurately mentions "including three proprietary models." Additionally, the citations for Claude-3-opus, Claude-3.5-sonnet, and Qwen2 among others are incorrect and urgently need correction. Concerning the topic categorization in Table 2, the paper does not provide a valid explanation or methodology for the classification. Several key steps in constructing the evaluation benchmarks are also lacking thorough explanations and supportive descriptions, even in the appendix. Overall, these issues raise concerns about the quality of the manuscript, and it is recommended that the authors give more attention and detailed exposition to these critical areas in the revision.

**Questions:**

1. Can the authors provide additional insights or data to illustrate the correlation between the model's performance on CodeMMLU and its real-world application in software development environments, where code generation is more prevalent?

2. In line 212, the authors claim to use a deep learning-based filtering model to automatically remove low-quality or irrelevant questions. I do not understand how the authors ensure data quality. Is it solely based on prompts? What model was used for checking? Are there any rule-based verification methods involved? I have reviewed sections A.2.1 and A.2.2 of the appendix and found no clear explanations. In line 1043, the authors state, “Followed by a manual validation step to ensure their appropriateness for the benchmark.” Did the authors really manually review over 10,000 samples? This seems hard to believe.

**Details Of Ethics Concerns:**

None.

---

> ### Author Response · Authors · 2024-11-23
>
> First of all, we would like to express our gratitude for your constructive feedback. We address your concerns or questions as follows.
>
> **Q1. Novelty concern**
> > [..] what issues CodeMMLU addresses that current benchmarks do not [..]
> >
>
> > [..] there are many comprehensive and thorough benchmarks for code understanding and code generation, such as CodeXGlue, XLCoST, xCodeEval, CodeScope, LiveCodeBench, and BigCodeBench. Notably, CodeScope (ACL 2024) has constructed a code understanding benchmark that includes four tasks and multiple-choice questions.
> >
>
> We appreciate the reviewer’s detailed feedback and suggestions regarding the Related Work section.
>
> CodeMMLU introduces a **multiple-choice question (MCQ) benchmark** that focuses on evaluating large language models (LLMs) on code understanding at scale, unlike prior benchmarks still rely on generation evaluation to assess code understanding (HumanEval, MBPP, BigCodeBench) and use match-based metrics such as BLEU, MRR, or ROUGE in tasks like code translation, code review, etc (CodeXGLUE, XLCoST, CodeScope).
>
> We argue that the MCQ format of CodeMMLU is critical in assessing the model’s code understanding at scale for two reasons. First, it is more straightforward and efficient to evaluate the MCQ’s answers compared to code generation or match-based metrics. CodeMMLU offers a large-scale evaluation that overcomes the scalability limitations inherent in execution-based metrics.
>
> To our knowledge, we are the first benchmark that attempts to bring multiple fundamental coding tasks, like code completion and code repair ***in the form of MCQs,*** to evaluate LLM. Thus, CodeMMLU focuses on both coding skills and evaluating programming knowledge understanding spread in diverse areas, making CodeMMLU the largest coding MCQ benchmark (with 20K questions, 52 topics, and 4 fundamental coding skill tests).
>
> Second, the data curation process to build CodeMMLU alleviates data leakages via several filtering steps and the usage of distractors. Furthermore, by swapping the ground truth position, the MCQ format engages the models in complex reasoning rather than simply memorizing the training data.
>
> We have revised sections 2, 3 in the main paper and appendices A.1 and B to highlight CodeMMLU’s novelty and discuss its contribution over existing benchmarks.
>
> **Q2. Data leakage issues**
> > LiveCodeBench addresses the issue of data leakage through dynamic data set updates. [..] (CodeMMLU) the paper does not offer any solutions to the data leakage problem mentioned in line 36.
> >
>
> We agree with the reviewer that data leakage is an important problem to consider when building a benchmark. Thus, we have taken extra efforts to alleviate this issue when building CodeMMLU. First, we employ several filtering processes to ensure data to be high quality. Then, the key contribution made in CodeMMLU is the reformulation of the raw data into the MCQ format (Section 3.2 and 3.3), which involves introducing synthetic distractors as incorrect answers. As a result, the questions in CodeMMLU are unlikely to be observed by the LLMs during training since they are more commonly trained on raw code, bug reports, etc. To quantify the data leakage degree of each benchmark, we follow BenBench [1] to report the perplexity and n-gram metrics.
>
> **Table 1.** `ppl` of benchmarks *(higher is better)*
>
> | **Models** | **CodeScope** | **CodeApex** | **CodeMMLU** |
> | --- | --- | --- | --- |
> | Mistral 7B v0.3 | 9.3152 | 16.0823 | **16.3178** |
> | DeepSeek Coder 7B v1.5 | 5.2571 | 9.3918 | **57.36** |
> | DeepSeek V2 Lite | 6.8899 | 11.987 | **1419.4829** |
> | Llama3.1 8B | 10.0514 | 123.2007 | **197.3058** |
>
> **Table 2.** `5-gram` of benchmarks *(lower is better)*
>
> | **Models** | **CodeScope** | **CodeApex** | **CodeMMLU** |
> | --- | --- | --- | --- |
> | Mistral 7B v0.3 | 0.25096 | 0.17025 | **0.13652** |
> | DeepSeek Coder 7b v1.5 | 0.28178 | 0.16804 | **0.14157** |
> | DeepSeek V2 Lite | 0.24925 | 0.15868 | **0.06866** |
> | Llama3.1 8B | 0.22185 | 0.13085 | **0.06523** |
>
> As shown in the following Table 1 and Table 2 (also Appendix A2 in the main paper), our CodeMMLU exhibits lower levels of data leakage (indicated by higher `ppl` and lower `n-gram`) than existing benchmarks such as CodeScope and CodeApex. This result demonstrates the effectiveness of CodeMMLU’s pre-processing pipeline in alleviating data leakage.

---

> > ### Author Response · Authors · 2024-11-23
> >
> > **Q3. Multiple-choice selection bias**
> > > [..] no strategies for eliminating this bias (the order option in multiple-choice question) are provided. This raises concerns about the reliability and robustness of benchmark tests. [..] the paper "Large Language Models Are Not Robust Multiple Choice Selectors," published at ICLR 2024, has demonstrated that using multiple-choice questions to evaluate LLMs is not stable and introduces significant biases.
> > >
> >
> > We appreciate the reviewer’s insightful question and the reference to the paper *"Large Language Models Are Not Robust Multiple Choice Selectors"* (PriDe) [2]. We recognize the sensitivity of LLMs to the order of options in MCQs, as highlighted both in our paper and prior work, including the referenced study. However, we respectfully disagree with the conclusion that MCQs are an unsuitable format for evaluating LLMs.
> >
> > The observed biases arising from option order are not inherent to the MCQ format itself but are indicative of limitations in the comprehension capabilities of current LLMs. Notably, humans do not experience a significant increase in difficulty due to changes in the order of options, underscoring that this is a model-specific issue rather than a fundamental flaw in MCQs. Moreover, MCQ-based benchmarks remain a widely accepted evaluation paradigm, as evidenced by their adoption in various prominent works [3,4]. Consequently, the biases reflect the areas where LLMs require further improvement rather than diminishing the reliability of our benchmark.
> >
> > **Table 3.** MCQ bias on MMLU (reported in [2])
> >
> > |  | **A** | **B** | **C** | **D** | **STD**  |
> > | --- | --- | --- | --- | --- | --- |
> > | Llama-30B | 68.2 | 54.1 | 50.1 | 41.2 | 9.74 |
> > | vicuna-v1.3-33B | 59.5 | 58.6 | 65.8 | 44.8 | 7.66 |
> > | Falcon-40B | 46.3 | 45.2 | 64.8 | 47.9 | 8.00 |
> > | Falcon-inst-40B | 38.8 | 38.9 | 55.7 | 69.1 | 12.69 |
> > | Llama-2-70B | 61.5 | 68.6 | 64.1 | 62 | **2.80** |
> > | Gpt-3.5-turbo | 65.3 | 68.5 | 74.2 | 60.9 | 4.85 |
> >
> > Furthermore, we provide the results highlighting that MCQs in the coding domain are more challenging than those in MMLU [5], as evidenced by a larger standard deviation introduced by several powerful LLMs. Table 4 compares to the reported result in [2] (Table 3). Interestingly, selection biases appear to diminish in more advanced models, such as GPT-4o, Claude 3.5 Sonnet, and Claude 3 Opus, suggesting that enhancing LLM robustness and consistency is key to mitigating these issues.
> >
> > **Table 4.** MCQ Bias on CodeMMLU. Accuracy standard deviation (STD) on order-changing experiment *(lower is better).*
> >
> > | Models | A | B | C | D | STD |
> > | --- | --- | --- | --- | --- | --- |
> > | GPT-4o | 80.49 | 78.05 | 71.34 | 70.12 | 4.38 |
> > | GPT-3.5-turbo | 51.22 | 43.29 | 47.56 | 54.88 | 4.30 |
> > | Claude3.5 Sonnet | 90.24 | 81.1 | 85.37 | 79.27 | 4.23 |
> > | Claude3.5 Haiku | 86.59 | 69.51 | 72.56 | 68.29 | 7.30 |
> > | Claude3 Opus | 79.27 | 77.44 | 82.32 | 84.76 | **2.81** |
> > | Claude3 Sonnet | 62.8 | 64.02 | 73.17 | 73.78 | 5.06 |
> > | Claude3 Haiku | 56.1 | 75 | 73.78 | 76.83 | 8.34 |
> > | Mixtral 8x7B | 22.56 | 74.39 | 71.95 | 63.41 | 20.91 |
> > | DSCoder 33B | 1.22 | 82.32 | 75.00 | 56.10 | 31.75 |
> > | DSCoder 7B | 40.85 | 74.39 | 64.02 | 39.02 | 15.10 |
> > | Phind-CL 34B | 6.10 | 90.85 | 75.00 | 46.34 | 32.21 |
> > | CL 34B Python | 0.61 | 77.44 | 70.73 | 49.39 | 30.09 |
> > | CL 34B Instruct | 9.15 | 84.76 | 65.24 | 46.34 | 27.91 |
> > | CL 13B Python | 0.61 | 54.88 | 70.12 | 12.20 | 28.85 |
> > | CL 13B Instruct | 2.44 | 68.29 | 72.56 | 29.88 | 28.85 |
> > | CL 7B Python | 0.00 | 90.24 | 14.02 | 0.61 | 37.39 |
> > | CL 7B Instruct | 3.66 | 1.22 | 93.90 | 15.85 | 38.07 |

---

> > > ### Author Response · Authors · 2024-11-23
> > >
> > > **Q4. Clarify the Data Creation Pipeline and Data Quality**
> > > > [..] the authors fail to clearly explain the process of constructing multiple-choice questions from various data forms after filtering.
> > > >
> > >
> > > > In line 319 of the text, although four models "GPT-3.5, GPT-4, Claude-3-opus, and Claude-3.5-sonnet" are tested, the manuscript inaccurately mentions "including three proprietary models."
> > > >
> > >
> > > > the citations for Claude-3-opus, Claude-3.5-sonnet, and Qwen2 among others are incorrect and urgently need correction.
> > > >
> > >
> > > > Several key steps in constructing the evaluation benchmarks are also lacking thorough explanations and supportive descriptions, even in the appendix.
> > > >
> > >
> > > Thank you for pointing out this issue. We have improved the clarity of the description for Figure 3 (now Figure 2 in the revised manuscript) and provided detailed explanations of the filtering methods in sections 3.3 and appendix A.2. These revisions aim to better illustrate the process of constructing multiple-choice questions. Additionally, we have clarified the role of LLM evaluations within the data creation pipeline. Please also refer to our response to Q5 for further details.
> > >
> > > We have addressed the specific concerns raised, including updating the correct number of evaluated models (lines 308, 346, 370), fixing incorrect citations, adding model descriptions (appendix C), and adding further details about the dataset construction process (appendix A.1), as outlined in our response above.
> > >
> > > **Q5. Provide Additional Insights on Correlation with Real-World Applications**
> > > > Can the authors provide additional insights or data to illustrate the correlation between the model's performance on CodeMMLU and its real-world application in software development environments, where code generation is more prevalent?
> > > >
> > >
> > > While real-world software engineering tasks may not explicitly present multiple-choice options, professionals often encounter implicit "MCQs" in their daily work, such as deciding between implementation strategies or choices of configuration. System-level coding or configuration [6] often involves selecting appropriate values from predefined options to configure the environment for various processes. For instance, when utilizing Hugging Face's Accelerate library to train large language models, developers must configure settings by selecting options such as MultiGPU or Single GPU usage, or choosing between training frameworks like DeepSpeed, FSDP or Megatron-LM. Furthermore, function calling has recently gained significant attention [7,8,9], requiring the selection of suitable libraries, frameworks, or tools for a specific task from a predefined set of options. These scenarios require similar decision-making processes to those evaluated in multiple-choice formats. Thus, MCQs can effectively distill these decisions into assessable components, bridging the gap between theoretical evaluation and practical application. However, measuring the correlation between CodeMMLU and these real-world application tasks in terms of model performance or data usage is beyond the scope of our current work and will be explored in future research.

---

> > > > ### Author Response · Authors · 2024-11-23
> > > >
> > > > **Q6. Concerning of Data Quality and Data filtering**
> > > > > In line 212, the authors claim to use a deep learning-based filtering model to automatically remove low-quality or irrelevant questions. I do not understand how the authors ensure data quality. [..]
> > > > >
> > > >
> > > > > In line 1043, the authors state, “Followed by a manual validation step to ensure their appropriateness for the benchmark.” Did the authors really manually review over 10,000 samples?
> > > > >
> > > >
> > > > We appreciate the reviewer’s concern and have clarified the data filtering process in Section 3.3 and Appendix A.1 in the revised manuscript. To ensure high data quality, we implemented a multi-step pipeline combining automated methods using rule-based approach and LLMs. First, rule-based techniques are adopted to remove data with incomplete information and non-textual. Second, three powerful LLMs (GPT-3.5, Mixture-8x7B, and Meta-LLaMA-3.1-8B-Instruct) were used to evaluate questions based on completeness, coherence, clarity, and relevance through prompting. We filtered out samples below a quality threshold. Third, a classification-based LLM categorized questions by topic and difficulty to ensure diversity and depth and filter out easy questions. For real-world coding tasks, we employ an execution-based filtering process to confirm the correctness of the correct options and the synthesized plausible incorrect ones. On the other hand, we manually review of a small subset of the data (100 instances per subject) in parallel of cleaning process to guarantee and update rule to ensure the effectiveness and quality of the automated filtering processes.
> > > >
> > > > ---
> > > > [1] Xu, R., Wang, Z., Fan, R. Z., & Liu, P. (2024). Benchmarking benchmark leakage in large language models. *arXiv preprint arXiv:2404.18824*.
> > > >
> > > > [2] Zheng, C., Zhou, H., Meng, F., Zhou, J., & Huang, M. (2023, September). Large language models are not robust multiple choice selectors. In *The Twelfth International Conference on Learning Representations*.
> > > >
> > > > [3] Wang, Y., Ma, X., Zhang, G., Ni, Y., Chandra, A., Guo, S., ... & Chen, W. (2024). Mmlu-pro: A more robust and challenging multi-task language understanding benchmark. *arXiv preprint arXiv:2406.01574*.
> > > >
> > > > [4] Ono, K., & Morita, A. (2024). Evaluating large language models: Chatgpt-4, mistral 8x7b, and google gemini benchmarked against mmlu. *Authorea Preprints*.
> > > >
> > > > [5] Hendrycks, D., Burns, C., Basart, S., Zou, A., Mazeika, M., Song, D., & Steinhardt, J. (2020). Measuring massive multitask language understanding. *arXiv preprint arXiv:2009.03300*.
> > > >
> > > > [6]  Kammakomati, M., Pimparkhede, S., Tamilselvam, S., Kumar, P., & Bhattacharyya, P. (2024). ConCodeEval: Evaluating Large Language Models for Code Constraints in Domain-Specific Languages. arXiv preprint arXiv:2407.03387.
> > > >
> > > > [7] Erdogan, L. E., Lee, N., Jha, S., Kim, S., Tabrizi, R., Moon, S., ... & Gholami, A. (2024). Tinyagent: Function calling at the edge. *arXiv preprint arXiv:2409.00608*.
> > > >
> > > > [8] Wang, X., Chen, Y., Yuan, L., Zhang, Y., Li, Y., Peng, H., & Ji, H. (2024). Executable code actions elicit better llm agents. *arXiv preprint arXiv:2402.01030*.
> > > >
> > > > [9] Zhang, J., Lan, T., Zhu, M., Liu, Z., Hoang, T., Kokane, S., ... & Xiong, C. (2024). xlam: A family of large action models to empower ai agent systems. *arXiv preprint arXiv:2409.03215*.

---

> ### Author Response · Authors · 2024-11-26
>
> Dear Reviewer **xQZ6**,
>
> Thank you for taking the time to provide your valuable comments. We hope our previous response has sufficiently addressed your concerns regarding the novelty of CodeMMLU, potential data leakage, and the reliability of MCQs against selection biases. We look forward to your feedback on our response and would be happy to clarify any additional questions.
>
> Best,
>
> Authors

---

> > ### Author Response · Authors · 2024-11-29
> >
> > Dear Reviewer **xQZ6**,
> >
> > We hope our previous response has adequately resolved your questions or concerns.
> > As the deadline for the ICLR rebuttal period is approaching, we look forward to hearing your feedback on our response, and would be pleased to clarify any additional questions.
> >
> > Best,
> >
> > Authors

---

### Author Response · Authors · 2024-11-25
**Response to All Reviewers**

We are deeply grateful to the reviewers for their detailed and constructive feedback. The insights have significantly enhanced the quality of our work. We are encouraged by reviewers xQZ6, aVYU, 6QYn, and wsRv, who recognized CodeMMLU's ability to evaluate LLMs across diverse programming topics at scale.

Reviewers 6QYn and wsRv highlighted our benchmark’s potential to reveal reasoning and comprehension weaknesses in LLMs, while reviewers xQZ6 and wsRv noted the robustness of our rigorous curation process and its potential to advance code understanding evaluations.

We address the key points raised by reviewers below:

- **Copyrighted concern and CodeMMLU licensing information (aVYU):**  We have clarified that the source data used for constructing CodeMMLU mostly comes from Common Crawl, as detailed in Appendix A3. For data crawled from websites such as W3Schools, GeeksforGeeks, and LeetCode, we fully comply with their copyrights or have sought their permissions to use such data for this project. CodeMMLU will be distributed under MIT License.
- **Potential data leakage (xQZ6, aVYU, wsRv):** We detailed our rigorous filtering processes and reformulated raw data into MCQs with synthetic distractors to minimize overlap with pretraining data, as described in Section 3.3 and Appendix A1. Furthermore, we added an experiment in Appendix A3, demonstrating that CodeMMLU exhibits lower leakage compared to other benchmarks using the benchmark benchmarking method provided by [1].
- **MCQ biases affecting the benchmark reliability (xQZ6, aVYU, wsRv):** To address concerns about MCQ biases, we conducted additional experiments in Appendix B2 using robust models from OpenAI and the Claude family, showing that weaker LLMs rely on memorization rather than comprehension. We believe this evidence will motivate improvements in LLM robustness and comprehension.
- **Quality of the filtering process (aVYU, 6QYn):** We detailed our multi-step pipeline in Section 3.3 and Appendix A1, including rule-based methods, LLM evaluations, and manual checks. We've also added a comprehensive explanation of our execution-based validation process that ensures distractor plausibility and semi-correctness.
- **CodeMMLU's contribution (xQZ6):** We have clarified how CodeMMLU uniquely evaluates large language models (LLMs) using a multiple-choice question (MCQ) format to assess comprehension at scale. This distinguishes it from prior benchmarks focused on generative tasks and provides new insights into LLM's comprehension capabilities in software development.

Based on the reviewers' suggestions, we have significantly improved the manuscript and clarified the raised concerns with supporting experiments. We hope this revision will address the reviewers’ concerns and strengthen the contributions of CodeMMLU as a reliable and scalable benchmark for evaluating LLMs in programming comprehension.

[1] Xu, R., Wang, Z., Fan, R. Z., & Liu, P. (2024). Benchmarking benchmark leakage in large language models. *arXiv preprint arXiv:2404.18824*.

---

### Meta-Review · Area_Chair_6Ufm · 2024-12-22

**Metareview:**

This paper introduces CodeMMLU, a new benchmark designed specifically to evaluate the code understanding capabilities of LLMs. The reviewers recognize the significance of the work and commend the comprehensive results and evaluation provided. However, some reviewers raised concerns about the writing, as well as the lack of comparisons or mentions of relevant prior work. The authors are encouraged to revise the paper in the final version to address these issues, particularly by improving the clarity and integrating comparisons with related works.

**Additional Comments On Reviewer Discussion:**

The reviewers have raised several questions, which have been properly addressed by the authors.

---

### Decision · Program_Chairs · 2025-01-22

Accept (Poster)